# Towards a New Standard: Prospective Validation of Ex Vivo Fusion Confocal Microscopy for Intraoperative Margin Assessment in Breast-Conserving Cancer Surgery

**DOI:** 10.3390/cancers17233848

**Published:** 2025-11-30

**Authors:** Daniel Humaran, Ana Castillo, Lidia Blay, Iciar Pascual, Karol Matute-Molina, Javiera Pérez-Anker, Susana Puig, Pedro L. Fernández, Joan F. Julián

**Affiliations:** 1Department of General and Digestive Surgery, Hospital Universitari Germans Trias I Pujol, Universitat Autònoma de Barcelona (UAB), 08916 Badalona, Spain; lydia.blay@gmail.com (L.B.); irizarp@gmail.com (I.P.); jfjulian.germanstrias@gencat.cat (J.F.J.); 2Department of Surgery, Universitat Autònoma de Barcelona (UAB), 08035 Barcelona, Spain; 3Department of Pathology, Institut de Recerca Germans Trias I Pujol, Hospital Universitari Germans Trias I Pujol, Universitat Autònoma de Barcelona (UAB), 08916 Badalona, Spain; acastillogan.germanstrias@gencat.cat (A.C.); karolmatutemolina@gmail.com (K.M.-M.); plfernandez.germanstrias@gencat.cat (P.L.F.); 4Department of Dermatology, Hospital Clínic de Barcelona, Universitat de Barcelona (UB), 08036 Barcelona, Spain; javiperezanker@gmail.com (J.P.-A.); susipuig@gmail.com (S.P.)

**Keywords:** breast-conserving surgery, intraoperative margin assessment, ex vivo fusion confocal microscopy, digital microscopy, breast cancer diagnosis

## Abstract

Breast cancer is the most common cancer in women, and breast-conserving surgery aims to remove the tumour while preserving breast appearance. A major challenge during this surgery is ensuring that no cancer cells remain at the edges of the removed tissue, as this can lead to recurrence and the need for additional surgery. Current intraoperative techniques for margin evaluation are often slow and technically demanding. This study explored a new real-time imaging method called ex vivo fusion confocal microscopy (EVFCM), which provides high-resolution visualisation of fresh tissue without complex preparation. In this prospective study, EVFCM demonstrated high diagnostic accuracy, reproducibility, and seamless integration into surgical workflows, enabling rapid assessment of tumour margins during surgery. By improving intraoperative decision-making and reducing the need for reoperations, EVFCM has the potential to enhance surgical precision, patient recovery, and the overall quality of care in breast cancer treatment.

## 1. Introduction

Breast cancer remains the most frequently diagnosed malignancy among women globally and continues to be a leading cause of cancer-related mortality [1]. Breast-conserving surgery has become the standard of care for early-stage breast cancer, providing oncological outcomes comparable to mastectomy while preserving breast aesthetics and improving patients’ quality of life [2], typically followed by adjuvant radiotherapy [3,4]. However, achieving histologically tumour-free resection margins remains a critical determinant of long-term success, as positive or close margins are associated with an increased risk of local recurrence [5,6,7,8]. In such cases, re-excision is often required, potentially delaying adjuvant therapy, compromising cosmetic outcomes, increasing healthcare costs, and contributing to psychological distress in patients [9,10,11,12,13,14].

Given the clinical importance of achieving clear margins during breast-conserving surgery, intraoperative margin assessment is considered a fundamental strategy to minimise the need for reoperation. Among the available techniques, frozen section analysis (FSA) is widely acknowledged as the gold standard microscopic method for rapid intraoperative margin evaluation, particularly in oncological procedures [15,16,17,18]. Despite its diagnostic potential, FSA is technically demanding, time-consuming, and dependent on specialised laboratory infrastructure and trained personnel [15,16,19]. Its reliability is further compromised in adipose-rich or fragmented breast tissue, where sectioning artefacts and sampling variability may distort the architecture and hinder both intraoperative and subsequent histopathological evaluations [18,20,21,22]. These factors, coupled with interpretative complexity and time constraints, may lead to false-negative results, ultimately limiting its diagnostic reliability of this method in routine clinical settings [23].

These challenges have resulted in reoperation rates exceeding 20% in international breast-conserving surgery series, highlighting the clinical need for improved intraoperative margin assessment strategies [18,24,25,26,27]. In this context, real-time digital pathology tools have attracted growing interest as potential solutions. Among these, ex vivo fusion confocal microscopy (EVFCM) has emerged as a promising approach [28,29,30], with the potential to distinguish between neoplastic and non-neoplastic lesions. This method offers real-time, high-resolution visualisation of fresh, unfixed tissue by combining reflectance and fluorescence channels to simulate the appearance of haematoxylin and eosin (H&E) staining [15,28,31,32,33]. Compared to FSA, EVFCM facilitates the rapid generation of digital scans, typically within minutes, requires fewer consumables and minimal tissue preparation, and can be deployed directly in the operating theatre, thereby eliminating the need to transport samples to the pathology department for evaluation [15]. Notably, this technique preserves the tissue integrity, allowing for subsequent evaluation using conventional H&E staining [34].

Acridine orange, a nucleic acid-selective fluorescent dye, is widely used in EVFCM protocols to enhance nuclear contrast in fresh breast tissue by binding to DNA and RNA without interfering with downstream processing. This enhancement facilitates the identification of cancer cells and improves the architectural definition [10].

Recent investigations have further highlighted the potential of confocal microscopy as a reliable tool for breast cancer diagnosis and intraoperative margin assessment. Several studies have reported high diagnostic accuracy when using confocal platforms for margin evaluation, demonstrating strong sensitivity, specificity, and inter-observer agreement in breast tissues [16,26,35,36]. The rapid acquisition of high-resolution images resembling conventional H&E histology has been consistently observed, confirming the value of EVFCM as a non-destructive, real-time imaging modality for surgical workflows. The HIBISCUSS trial has further underscored its clinical feasibility, validating its use by both surgeons and pathologists in the intraoperative setting [26].

Despite these features and encouraging results, the clinical translation of EVFCM remains limited. This is largely attributable to the methodological heterogeneity of existing protocols, which vary in terms of staining procedures, evaluator training, imaging workflows, and interpretation criteria. Such inconsistencies have hindered reproducibility, limiting the generalisability of findings across different settings and highlighting the need for standardised approaches and robust prospective validation. Importantly, EVFCM does not require a complex laboratory infrastructure and can be implemented directly in the surgical setting. Moreover, recent studies suggest that the learning curve associated with image interpretation is relatively short [33,37], further supporting its feasibility for intraoperative application. While several technologies compete for intraoperative margin assessment, the role of EVFCM as a histologically faithful, real-time, and tissue-preserving alternative remains to be fully elucidated.

To address these limitations, this study was designed with the objective of prospectively validating a reproducible protocol for EVFCM applied to intraoperative margin assessment in breast-conserving surgery. By combining a rapid dual-staining method with blinded evaluation by experienced breast pathologists, we sought to determine the diagnostic accuracy of EVFCM compared with that of conventional H&E histology. We also aimed to assess its clinical feasibility, reproducibility, and diagnostic performance in terms of sensitivity, specificity, and predictive value. Ultimately, this study aspires to support the integration of EVFCM into routine surgical workflows, with the potential to reduce reoperation rates and optimise patient outcomes.

## 2. Materials and Methods

### 2.1. Study Design and Setting

A prospective observational diagnostic validation study was conducted to evaluate the utility of EVFCM as a tool for the intraoperative assessment of surgical margins in breast-conserving surgery for breast cancer. Samples were processed in the immediate postoperative setting, allowing for a direct comparison between EVFCM findings and conventional histopathological analysis, which served as the diagnostic gold standard.

Sample collection was performed consecutively between October 2023 and July 2025 at the Hospital Universitari Germans Trias i Pujol (Badalona, Spain), resulting in a total of 146 breast tissue samples being obtained from 98 patients. Among these, 44 were main tumorectomy specimens, and 102 were cavity shave margins enlargements. In accordance with STARD principles, each specimen was considered an independent diagnostic unit, as no patient contributed multiple samples from the same anatomical margin or repeated resections of the same area. This ensured that every EVFCM–histology comparison represented a distinct diagnostic event and prevented hierarchical clustering that could otherwise bias diagnostic performance metrics. The inclusion criteria comprised breast tissue specimens with a confirmed diagnosis of breast cancer, resected during breast-conserving surgery, clearly identifiable post-resection, and of sufficient volume to allow parallel analysis via EVFCM and conventional histology. The exclusion criteria were samples from patients who had received neoadjuvant chemotherapy or specimens of insufficient size, where manipulation could compromise histopathological evaluation.

### 2.2. Surgical Procedure

All patients underwent breast-conserving surgery performed by the dedicated breast surgery team of our centre, in accordance with institutional protocols and international oncological standards.

Immediately after excision, the tumorectomy specimens were oriented by the operating surgeon using sutures, clips, and inked margins, following the institutional margin mapping protocol to ensure precise spatial correlation during pathological evaluation.

Macroscopic inspection and orientation were jointly performed by the breast surgeon and the pathologist immediately after excision. This collaboration ensured that the areas most suspicious for residual disease were selected for EVFCM imaging, aligning macroscopic judgement with microscopic evaluation. In our workflow, EVFCM assessment was integrated at this same decision point, functionally replacing frozen section analysis.

For perioperative margin evaluation with EVFCM, a targeted approach was applied to areas considered to be at the highest risk for residual disease, guided by macroscopic inspection and surgical judgement. Two types of specimens were analysed depending on the surgical context:Main tumorectomy specimens: For standard tumour excisions, the surface region considered by the surgeon and pathologist closest to the lesion was selected for confocal imaging, targeting the area most likely to harbour residual tumour cells.Cavity shave margins: It is important to note that cavity shave margins were not routinely performed in all patients but were selectively excised in cases where the operating surgeon considered additional peripheral tissue removal clinically justified. This selective use reflects standard intraoperative judgement in our institution and does not define two distinct surgical populations. In selected cases, additional margins were excised to widen the resection around the tumour bed. To preserve orientation, the surface of each re-excision specimen that had been in contact with the main tumour specimen was marked with blue ink immediately after removal. The opposite surface, representing the interface with the remaining breast tissue, was then analysed using confocal imaging to assess potential residual disease at the preserved margin.

This dual-sample strategy enabled the assessment of both the oncological adequacy of the primary excision and the involvement of peripheral margins, potentially requiring further resection.

For each specimen, only one surface was sampled for EVFCM assessment. No specimen underwent repeated sampling of the same anatomical area, and no re-excisions of an already sampled margin were performed. All remaining surfaces of the specimen were subsequently evaluated on definitive paraffin-embedded histology, and no case demonstrated a positive margin on a surface different from the one selected for EVFCM. Consequently, no hidden false negatives occurred due to sampling selection, and each EVFCM–histology comparison corresponded to a single, unique diagnostic event.

Following resection, specimens and margins were delivered fresh to the adjacent Department of Pathology, where EVFCM imaging was performed, preserving tissue integrity. In our centre, the device is located immediately next to the surgical suite, allowing specimen transfer within approximately one minute. As a result, EVFCM analysis was carried out in the immediate perioperative setting, not physically inside the operating theatre. The surgical and pathology teams collaboratively prepared the specimens to ensure accurate orientation and imaging reliability.

### 2.3. Staining Protocol, Image Acquisition, and Histopathological Processing

Samples were stained using a rapid protocol optimised to enhance the contrast of the nuclei, cytoplasm, and extracellular matrix. Each specimen was first pre-treated with 70% ethanol for 10 s to induce superficial dehydration and protein precipitation, thereby improving tissue permeability and optical contrast [38]. Sequential staining was then performed using acridine orange (Sigma-Aldrich, MerckKGaA, Madrid, Spain) for 30 s and fast green FCF (Sigma-Aldrich, MerckKGaA, Madrid, Spain) for 20 s.

Acridine orange is a fluorescent dye that binds specifically to nucleic acids and emits green fluorescence upon excitation at 488 nm, enabling clear visualisation of nuclear structures in confocal images [10]. The emitted fluorescence of acridine orange was collected above 500 nm. Fast green FCF stains the cytoplasm and extracellular matrix, enhancing the structural details, including fibrotic areas [39], contributing only to reflectance-based contrast and was not excited as a fluorescent dye. The combined use of these dyes enhances the architectural definition of fresh breast tissue, without compromising sample integrity.

Images were acquired using a VivaScope 2500M-G4 ex vivo confocal microscope (Mavig GmbH, Munich, Germany; Caliber I.D., Rochester, NY, USA) [40]. This device is equipped with a 38×, 0.85 numerical aperture (NA) water-immersion objective with spherical and chromatic aberration correction optimised for en-face confocal imaging of fresh tissue. This optical configuration provides a lateral optical resolution of <1.25 µm and an axial resolution of <5 µm at the centre of the field of view. This system employs two simultaneous laser channels—fluorescence (488 nm) with excitation of acridine orange and reflectance (638 nm)—which are automatically merged using the proprietary acquisition software (software version 4.2) to produce high-resolution fused confocal images.

All EVFCM acquisitions were performed using the manufacturer’s preset for fresh ex vivo tissue, which automatically adjusts laser power, detector gain, pixel dwell time, and sampling density to optimise image contrast and resolution. These parameters are factory-calibrated, not user-modifiable, and ensure consistency and reproducibility across samples. Pixel sampling corresponds to the system’s native optical resolution and complies with Nyquist sampling criteria, providing sufficient detail for accurate assessment of nuclear and architectural morphology.

The system enables optical focusing of fresh tissue up to a depth of approximately 200 µm, with an optimal image resolution and signal-to-noise ratio achieved within the superficial 100–150 µm layer of the tissue. This depth range is suitable for the intraoperative assessment of surface margins, enabling the detection of tumour cells at or near the resection interface. Confocal imaging was performed in en-face mode, generating two-dimensional optical sections from the superficial tissue layer as a single optical plane. For each specimen, the focal plane used for analysis corresponded to the depth providing maximal resolution of fluorescence signal and cellular contrast. No volumetric (z-stack) acquisition, projection methods, or post-acquisition processing were applied.

Large-area confocal mosaics were obtained using the system’s automated mosaic-acquisition function, which captures adjacent fields of view and merges them through the system’s proprietary software (software version 4.2) to produce a continuous en-face image. The manufacturer does not disclose tile overlap percentages or field-of-view dimensions for individual tiles; these values are internally calibrated and verified during quality-control procedures to ensure alignment accuracy. All specimens were imaged under identical automated settings.

The images were visualised using the VivaScope onboard proprietary software (software version 4.2), which automatically applies the system’s integrated pseudocoloring algorithm to merge the 488 nm fluorescence and 638 nm reflectance channels into a single composite image that simulates H&E staining. The images analysed represent the direct output generated by the system, with no external software or additional post-processing applied.

Each scan covered an area of approximately 3 × 3 cm. The EVFCM processing phase, defined as staining, mounting, and image acquisition once the specimen had been received and oriented in the pathology laboratory, was consistently completed in under 5 min per sample.

Images were stored and exported in 8-bit TIFF format while preserving the full spatial resolution of the acquisition as single-plane files with pseudocoloring automatically generated by the device software (software version 4.2), and no external conversion or bit-depth modification was performed to ensure traceability and support a structured diagnostic review.

Following EVFCM imaging, all samples underwent routine formalin fixation, paraffin embedding, sectioning, and staining with H&E to establish a definitive diagnosis. No washing or destaining step was required. Acridine Orange and Fast Green FCF are applied superficially and exhibit minimal tissue penetration. Both dyes are removed during routine fixation, dehydration, and paraffin embedding, without affecting subsequent H&E staining or histopathological interpretation.

These histological sections served as the reference standard for comparison with the EVFCM findings.

H&E-stained slides were digitised using the departmental whole-slide scanner PANNORAMIC 100 flash DX (3DHISTECH Lts, Budapest, Hungary) (P1000) at 20× magnification (0.24 μm/pixel), exported in MRXS format, and reviewed using ClinicalViewer (3DHISTECH Ltd., Budapest, Hungary). No additional post-processing was performed.

### 2.4. Image Evaluation

Sample preparation and EVFCM processing were performed either by a breast surgeon with pathology training in close collaboration with breast pathologists or by the breast pathologists trained in EVFCM, following the same standardised workflow. This ensured consistent handling of all specimens, correct orientation of the selected tissue surface, and uniform application of the staining and imaging protocol across all cases.

Diagnostic image evaluation was independently performed by two expert breast pathologists experienced in EVFCM image interpretation. All images were independently and blindly assessed without access to the corresponding definitive histology, ensuring an unbiased assessment.

Subsequently, paraffin-embedded sections were evaluated by the same two pathologists after a wash-out period of at least two months to minimise recall bias. Any discrepancies between the pathologists were resolved through a consensus review to establish the final diagnostic classification, which was considered the gold standard.

The evaluation also included an assessment of imaging artefacts and suitability. An artefact was defined as present when more than 10% of the tissue could not be evaluated due to staining, processing, or technical issues. Image quality was considered unsuitable if less than 80% of the sample was reliably evaluated. According to these criteria, two specimens were excluded from the final analysis because their EVFCM images were not reliably assessable (one due to motion artefact and one due to incomplete staining), resulting in less than 80% evaluable tissue area.

### 2.5. Statistical Analysis

Each specimen was considered an independent sample for diagnostic comparison. An exploratory data analysis was performed using frequency and percentage distribution. To evaluate the diagnostic performance of EVFCM compared with the definitive histopathological diagnosis on H&E-stained sections (gold standard), sensitivity, specificity, positive predictive value (PPV), negative predictive value (NPV), and overall accuracy were calculated. For multi-class variables, a one-versus-all approach was adopted.

The agreement between EVFCM and conventional histology, as well as the inter-observer agreement between pathologists, was assessed using Cohen’s κ coefficient with 95% confidence intervals obtained through bootstrap resampling. Kappa values were interpreted according to the Landis and Koch scale, with values > 0.80 considered to indicate almost perfect agreement. For binary outcomes, McNemar’s test was used to assess the marginal homogeneity. Statistical significance was set at a *p*-value < 0.05.

All data processing and statistical analyses were performed using R software (version 4.4.1; R Foundation for Statistical Computing, Vienna, Austria).

### 2.6. Ethical Considerations

This study was approved by the Clinical Research Ethics Committee of the Hospital Universitari Germans Trias i Pujol and was conducted in accordance with the ethical principles of the Declaration of Helsinki (MCF-CM, PI-16-161, 07/11/2016). All patients provided informed consent, authorising the processing of their samples for biomedical research purposes. Sample handling adhered to established protocols regarding confidentiality, custody, and traceability.

## 3. Results

### 3.1. Study Population and Case Flow

A total of 146 consecutive breast specimens were prospectively collected, of which 144 were evaluable for EVFCM analysis and comparison to definitive histology (Figure 1 and Figure 2). Two samples were excluded because less than 80% of the tissue area was reliably evaluable, and they were therefore excluded from the diagnostic analysis. The overall inclusion and exclusion process is summarised in Figure 3, which follows the STARD guidelines for diagnostic accuracy studies.

### 3.2. Inter-Observer Agreement Between Pathologists

The inter-observer agreement between the two pathologists is summarised in Table 1. Across the 146 paired evaluations, the agreement for the presence of neoplasia was almost perfect, with 97.3% concordance and a Cohen’s κ of 0.942 (95% CI: 0.881–0.986). Tumour type classification (invasive, in situ, or no tumour) also showed very high agreement (93.8%, κ = 0.883, 95% CI: 0.807–0.948), as did the classification of invasive subtype (ductal, lobular, or other), with 97.3% concordance (κ = 0.946, 95% CI: 0.885–0.987). Lower agreement was observed for artefact recognition (84.2% concordance, κ = 0.355, 95% CI: 0.143–0.547), corresponding to fair agreement, while specimen suitability exhibited substantial agreement (98.6%, κ = 0.651, 95% CI: 0.535–0.767). These findings indicate excellent reproducibility in the assessment of key diagnostic parameters, with minor variability limited to artefact recognition and suitability assessment.

### 3.3. Diagnostic Performance of EVFCM Compared with Paraffin Histology

The diagnostic performance metrics for EVFCM consensus diagnosis compared with paraffin-embedded histology are presented in Table 2, which includes 144 evaluable cases after excluding two unsuitable samples by consensus.

For tumour detection, EVFCM achieved a sensitivity of 93.7% and specificity of 93.7% (κ = 0.929, 95% CI: 0.861–0.986, *p* < 0.001). Tumour type classification (no tumour, invasive, in situ, or invasive + in situ) showed an overall accuracy of 95.8% (κ = 0.925, 95% CI: 0.859–0.975, *p* < 0.001). For invasive subtype identification (ductal, lobular, or other), accuracy reached 95.1% (κ = 0.907, 95% CI: 0.831–0.970, *p* < 0.001), with a slightly lower sensitivity observed for lobular carcinoma compared with the other subtypes.

Margin assessment showed a sensitivity of 80.0%, specificity of 100.0%, and overall accuracy of 99.3% (κ = 0.857, 95% CI: 0.566–1.000, *p* < 0.001). Evaluation of margin distance (<2 mm vs. ≥2 mm) resulted in a sensitivity of 75.0%, specificity of 100.0%, and κ = 0.854 (95% CI: 0.566–1.000, *p* < 0.001). Overall, EVFCM demonstrated a high level of diagnostic concordance with conventional histology, confirming its reliability for the accurate intraoperative assessment of tumour presence, type, and margin status.

## 4. Discussion

Building upon our previous experience with ex vivo fusion confocal microscopy (EVFCM) [37] and the expanding body of evidence supporting its diagnostic value, this prospective validation study aimed to consolidate its clinical applicability in breast-conserving surgery. The primary objective was to confirm the accuracy of EVFCM in detecting breast neoplasia, classifying tumour subtypes, and assessing surgical margins in real time, using conventional paraffin-embedded histology as the gold standard. The direct correlation between EVFCM images and their corresponding H&E sections enabled a precise morphological comparison, allowing for a robust assessment of diagnostic performance and reproducibility.

Across 144 evaluable cases, EVFCM demonstrated high sensitivity, specificity, and overall diagnostic accuracy for both tumour detection and margin evaluation, supported by almost perfect interobserver agreement between breast pathologists. These findings support the reproducibility and reliability of this imaging technique and support its potential integration into the intraoperative workflow. Furthermore, tissue integrity was preserved throughout the EVFCM process, enabling complementary histopathological verification without compromising the standard pathological evaluation.

The diagnostic performance observed in this study corroborates the capacity of EVFCM to reproduce conventional histological findings with a high fidelity. Sensitivity and specificity values exceeding 90% for tumour detection highlight the reliability of this method in distinguishing malignant from non-malignant breast tissue. These results are in close agreement with those reported by Conversano et al. [26] in the multicentric HIBISCUSS study, which demonstrated the high diagnostic accuracy and reproducibility of confocal microscopy in breast tissue imaging, and are consistent with the findings of a recent systematic review by Au et al. [15], which confirmed the strong diagnostic performance of EVFCM across various solid tumour types. Colard-Thomas et al. [18] confirmed its feasibility for intraoperative margin assessment within the pathology workflow. Collectively, these data reinforce the reproducibility and growing clinical validation of confocal-based imaging for real-time breast tissue evaluation across independent research groups.

Inter-observer agreement between pathologists was also excellent, with κ values exceeding 0.88 for both neoplasia identification and tumour typing. This emphasises the consistency of EVFCM interpretation among trained evaluators and validates the standardisation achieved through the proposed imaging and staining protocol. The only minor variability occurred in artefact recognition and suitability assessment, likely reflecting the subjective perception of image quality rather than true diagnostic disagreement. Importantly, this variability did not affect the diagnostic classification or accuracy outcomes, further supporting the robustness of the EVFCM as a consistent diagnostic tool. This variability in artefact identification also underscores an important methodological aspect: artefact recognition in fresh-tissue confocal microscopy remains partially subjective, as unified and universally accepted criteria are not yet established. Differences in individual thresholds for defining evaluable versus non-evaluable regions, as well as varying familiarity with confocal artefacts, may account for the discrepancies observed in Table 1. Rather than weakening the method, this observation strengthens the rationale for digital approaches, which provide standardised image acquisition, minimise operator-dependent variability, and offer future potential for computational support and automated artefact detection within digital pathology workflows.

Regarding invasive subtypes, EVFCM demonstrated high overall diagnostic accuracy and almost perfect agreement with paraffin histology findings. Lower sensitivity (40%) was observed for invasive lobular carcinoma compared with ductal and mixed subtypes, which aligns with its well-recognised histopathological features characterised by reduced cellular cohesion, a diffuse growth pattern, and minimal architectural distortion [10]. Such morphological attributes often make their recognition more challenging, not only in conventional histology but also in confocal imaging, where tumour boundaries may appear less sharply defined. This observation further supports the need for tailored interpretative training for these less cohesive tumour subtypes. However, given the small number of ILC cases included in our cohort, this sensitivity value should be interpreted as preliminary rather than conclusive. Larger and more balanced datasets are required to establish robust subtype-specific performance metrics and validate whether this trend persists in broader clinical settings.

Margin assessment remains a critical determinant of oncological safety in breast-conserving surgery, directly influencing re-excision rates and long-term local control. In this study, the EVFCM achieved a sensitivity of 80%, specificity of 100%, and an overall diagnostic accuracy of 99.3% for margin detection (κ = 0.857, *p* < 0.001). Four of the five histologically involved margin cases were correctly identified using EVFCM. Although the number of positive margin cases is small, these preliminary findings are encouraging and support the potential of EVFCM for intraoperative margin assessment, warranting confirmation in larger series. A targeted review of discordant cases revealed that the single misclassified margin corresponded to ductal carcinoma in situ (DCIS) associated with epithelial hyperplasia near the resection edge. The morphological overlap between these entities likely contributes to misclassification, representing a recognised diagnostic challenge in both frozen and optical-based intraoperative techniques. Comparable diagnostic challenges have been reported by Sandor et al. [35], who noted that the identification of DCIS lesions in lumpectomy margins remains one of the main challenges in breast-conserving surgery and that distinguishing them from benign epithelial proliferations or low-grade lesions may be difficult due to the complex and heterogeneous architecture of DCIS, which can further complicate its intraoperative recognition in confocal images. Importantly, this isolated discrepancy did not affect the overall diagnostic reliability, supporting that EVFCM is a robust method for intraoperative margin assessment.

Nevertheless, the sensitivity estimate for margin positivity is based on only five cases and should therefore be considered preliminary. Moreover, reoperation rates reflect multifactorial clinical outcomes and cannot be directly compared with diagnostic sensitivity. Additionally, this present study was not designed to evaluate the impact of EVFCM on reoperation rates—an outcome influenced by multiple surgical, pathological, and institutional factors beyond intraoperative imaging performance.

Beyond the binary margin status, the capacity of EVFCM to assess the distance between tumour cells and the resection edge represents an additional diagnostic advantage. In this study, the evaluation of margin distance (<2 mm vs. ≥2 mm) yielded a sensitivity of 75% and specificity of 100% (κ = 0.854, *p* < 0.001). These results demonstrate that EVFCM can not only detect residual tumours but can also estimate their spatial proximity to the margin, providing valuable information for intraoperative decision-making. Given that the local recurrence risk increases as the tumour distance to the margin decreases, the ability to identify cases within this critical threshold (<2 mm) offers significant prognostic value. This quantitative capability enhances the clinical utility of EVFCM, enabling surgeons to perform targeted re-excisions during the same procedure, thereby reducing the likelihood of secondary surgeries. This feature is particularly relevant in borderline cases, where intraoperative estimation of tumour margin distance can guide immediate surgical extension and improve oncological safety.

Our results place EVFCM among the most accurate optical platforms for intraoperative margin evaluation, with a diagnostic performance comparable to that reported in the literature. Previous studies have documented sensitivities between 75% and 95% and specificities up to 98% [15,26,36,41], while the recent SHIELD trial demonstrated similar values (80.9% sensitivity and 99.5% specificity) together with a reduction in re-excision rates [41]. These converging data support that EVFCM achieves high diagnostic precision while maintaining workflow efficiency, reinforcing its potential for routine intraoperative implementation during breast surgery.

Nevertheless, when comparing our findings with those reported in studies using the Histolog^®^ Scanner—such as Togawa et al. [36] and the SHIELD trial by Lux et al. [41]—it is important to acknowledge the substantial methodological differences between platforms. The Histolog^®^ Scanner is a large field-of-view confocal laser scanning microscope designed for rapid macroscopic en-face imaging and is used intraoperatively by breast surgeons following a single-step acridine-orange stain. Its approximate lateral resolution of 2 μm is inherently lower than that of VivaScope 2500M-G4, which produces higher-resolution fused reflectance–fluorescence images using a dual-contrast acridine-orange–fast-green protocol optimised for enhanced nuclear and stromal detail. Furthermore, while Histolog^®^ assessments were performed intraoperatively by surgeons (exclusively so in the SHIELD trial), EVFCM imaging in our study was performed ex vivo and interpreted by trained breast pathologists within a controlled diagnostic workflow. These differences in device design, optical resolution, staining chemistry, and operator expertise inevitably influence diagnostic performance across studies.

Accordingly, comparisons between Histolog^®^-based intraoperative assessments and EVFCM should be interpreted within the context of their distinct optical principles and intended clinical applications. Rather than being directly comparable, the two systems represent complementary approaches positioned at different points along the intraoperative–pathology continuum, with EVFCM offering higher-resolution tissue characterisation and potential for future intraoperative integration as workflows evolve to support ex vivo high-resolution assessment.

The excellent diagnostic accuracy and reproducibility observed here, coupled with a scanning time of less than five minutes, indicate that EVFCM could be feasibly incorporated into the intraoperative workflow without extending the surgical time. The ability to image fresh tissue directly could potentially substitute the need for frozen sectioning, reduce artefacts, and preserve samples for subsequent histopathological analysis. The high concordance with paraffin histology further validates EVFCM as a reliable real-time imaging tool capable of supporting intraoperative decision-making and providing immediate feedback to the surgical team.

The workflow proposed in this study also demonstrates the practical feasibility of EVFCM in surgical settings and represents one of its major strengths. Staining and image acquisition were completed in less than five minutes, ensuring compatibility with surgical timescales. This method facilitates the real-time acquisition of histology-like digital images, allowing pathologists and surgeons to assess margin status without interrupting the operative sequence. Beyond its technical performance, EVFCM offers several logistical advantages over FSA: it requires no cryostat, minimises tissue loss, preserves samples for definitive diagnosis, and can be deployed adjacent to the operating theatre, reducing specimen transport time and preserving orientation accuracy. Therefore, it can be evaluated either on-site or remotely.

These advantages facilitate rapid and continuous feedback between surgeons and pathologists, enabling a genuine, real-time diagnostic dialogue during surgery. The immediate visualisation of confocal images allows direct communication and joint interpretation of findings, effectively integrating both perspectives into intraoperative decision-making. This multidisciplinary and close interaction between the surgical and pathology teams constitutes one of the key operational strengths of the EVFCM, supporting informed and timely margin management without disrupting the operative sequence. The high diagnostic reproducibility achieved across evaluators supports the use of EVFCM as a reliable intraoperative adjunct for margin assessment in breast-conserving surgery. With appropriate training and standardised protocols, EVFCM could represent a sustainable and accessible alternative to FSA, particularly in centres with limited histopathological infrastructure.

This study has certain limitations. Although the sample size enabled a robust statistical analysis, the data were obtained from a single institution, which may limit external generalisability. Additionally, the number of positive margin cases in our cohort was relatively small, reflecting the naturally low rate of margin involvement achieved by our specialised breast surgery team. While this limited representation may have constrained the precision of sensitivity estimates for margin positivity, it simultaneously underscores the need for larger, multicentric validation studies to confirm reproducibility across different workflow settings, staining protocols, and broader case distributions. Although image interpretation followed standardised training, EVFCM evaluation inherently involves a short learning curve, particularly for subtle histological variants and morphological patterns such as invasive lobular carcinoma. Therefore, future studies should explore optimised training models and interobserver calibration strategies to further enhance diagnostic consistency. In addition, further multicentric research should specifically address cases known to pose particular diagnostic challenges, including ductal carcinoma in situ (DCIS), multifocal disease, and narrow margins (<2 mm). These scenarios often present complex or subtle morphological patterns that can hinder intraoperative recognition, even in conventional histopathology. Expanding the validation of EVFCM across these subgroups will be essential to confirm its reproducibility, refine interpretative criteria, and ensure its applicability in all clinically relevant contexts of breast-conserving surgery. Finally, the optical penetration depth of the device may preclude the detection of deeper submarginal lesions. Nonetheless, as intraoperative assessment primarily focuses on surface margins, this limitation has limited clinical relevance when the method is applied correctly. Despite these limitations, the strong diagnostic consistency and high reproducibility observed across the evaluators reinforce the robustness of our findings.

Looking ahead, the integration of artificial intelligence (AI) and machine learning algorithms into EVFCM workflows holds great potential for automated tumour detection, potentially reducing interobserver variability and expediting intraoperative reporting. A recent systematic review by Dur Karasayar et al. [42] highlighted how AI-driven digital pathology enhances diagnostic precision, grading reproducibility, and biomarker quantification in breast cancer, paving the way for more standardised and efficient histopathological workflows. Extending such advances to the EVFCM could enable automated tissue classification and objective intraoperative interpretation, thereby streamlining real-time decision-making. Additionally, the incorporation of EVFCM into digital pathology networks could facilitate remote consultations and real-time multidisciplinary collaborations, expanding accessibility and scalability across institutions.

In summary, this study provides strong evidence supporting the diagnostic reliability and clinical feasibility of EVFCM for intraoperative margin assessment in breast-conserving surgery. Through a standardised workflow and dual evaluation by expert breast pathologists, EVFCM achieved high diagnostic accuracy and reproducibility across all key parameters. Its near-perfect agreement with paraffin histology and rapid image acquisition underscores its potential as a histology-like intraoperative tool. By combining speed, accuracy, and tissue preservation, EVFCM bridges the gap between intraoperative decision-making and definitive histopathological diagnosis. Its implementation could contribute to reducing re-excision rates, optimising surgical precision, and ultimately improving patient outcomes. These findings establish a solid foundation for future multicentric validation and digital integration studies aimed at consolidating EVFCM as a new standard in intraoperative breast pathology, bridging technology and surgical precision to improve oncological safety and patient care. As intraoperative imaging continues to evolve, EVFCM stands out as a promising bridge between precision surgery and real-time pathology with the potential to redefine the standards of breast-conserving surgery.

## 5. Conclusions

This prospective validation study successfully met its primary objectives, demonstrating that ex vivo fusion confocal microscopy (EVFCM) is a reliable, accurate, and clinically feasible method for intraoperative margin assessment in breast-conserving surgery. The EVFCM achieved high sensitivity, specificity, and near-perfect interobserver agreement, confirming its diagnostic reproducibility and ability to faithfully reproduce histological architecture in fresh, unfixed tissue.

Through a rapid and standardised workflow, the EVFCM processing phase, defined as staining, mounting, and image acquisition once the specimen has already been oriented and received in the pathology laboratory, was consistently completed in under five minutes. EVFCM enables real-time evaluation of tumour presence, margin status, and tumour margin distance without compromising tissue integrity. This additional quantitative capability provides significant intraoperative value by supporting immediate surgical decisions in borderline cases, thereby potentially enhancing the oncological safety.

The simplicity of its implementation, combined with its high diagnostic precision, positions EVFCM as a practical and sustainable alternative to frozen section analysis, particularly in settings with limited histopathological infrastructure. Looking ahead, the integration of EVFCM into digital pathology platforms and artificial intelligence systems could further streamline image interpretation, enable remote collaboration, and expand accessibility across institutions.

Beyond its technical and diagnostic advantages, the widespread adoption of EVFCM could yield tangible benefits for patients by helping to reduce re-excisions, preserving cosmetic outcomes, and alleviating the psychological and physical burden of repeated surgeries. By fulfilling its diagnostic and clinical objectives, this study consolidates EVFCM as a robust intraoperative imaging tool that bridges technology and surgical precision, paving the way for its integration into future surgical standards and improved patient care.

## Figures and Tables

**Figure 1 cancers-17-03848-f001:**
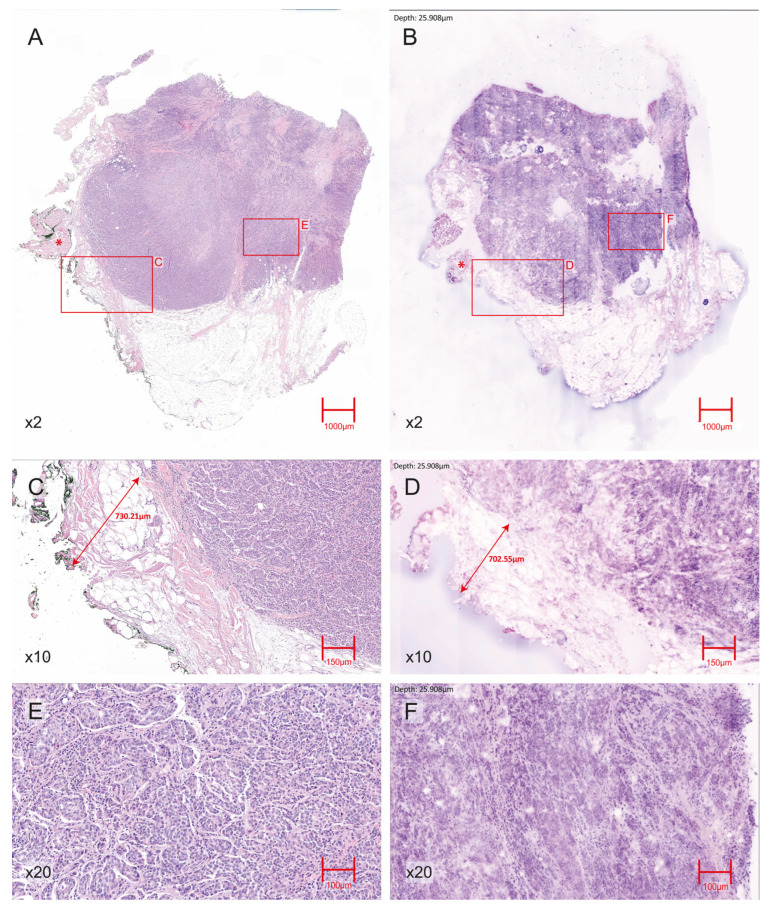
Representative H&E- EVFCM correlation for an invasive ductal carcinoma (IDC) with a negative surgical margin. (**A**) H&E section (2× magnification) showing the IDC within the breast parenchyma; an asterisk (*) indicates adjacent skeletal muscle fibres. (**B**) Corresponding 2× magnification EVFCM image of the same region, reproducing overall tumour architecture and showing the same skeletal muscle area (*). (**C**) H&E section (10× magnification) view of the tumour–margin interface, with scale annotation indicating a tumour–margin distance of approximately 7 mm, confirming the absence of tumour cells at the resection edge. (**D**) Corresponding EVFCM image (10× magnification) of the same interface with equivalent scale annotation, consistently demonstrating a negative margin. (**E**) H&E section (20× magnification) showing nuclear pleomorphism and the infiltrative growth pattern characteristic of IDC. (**F**) 20× magnification EVFCM image illustrating comparable cytological and architectural features.

**Figure 2 cancers-17-03848-f002:**
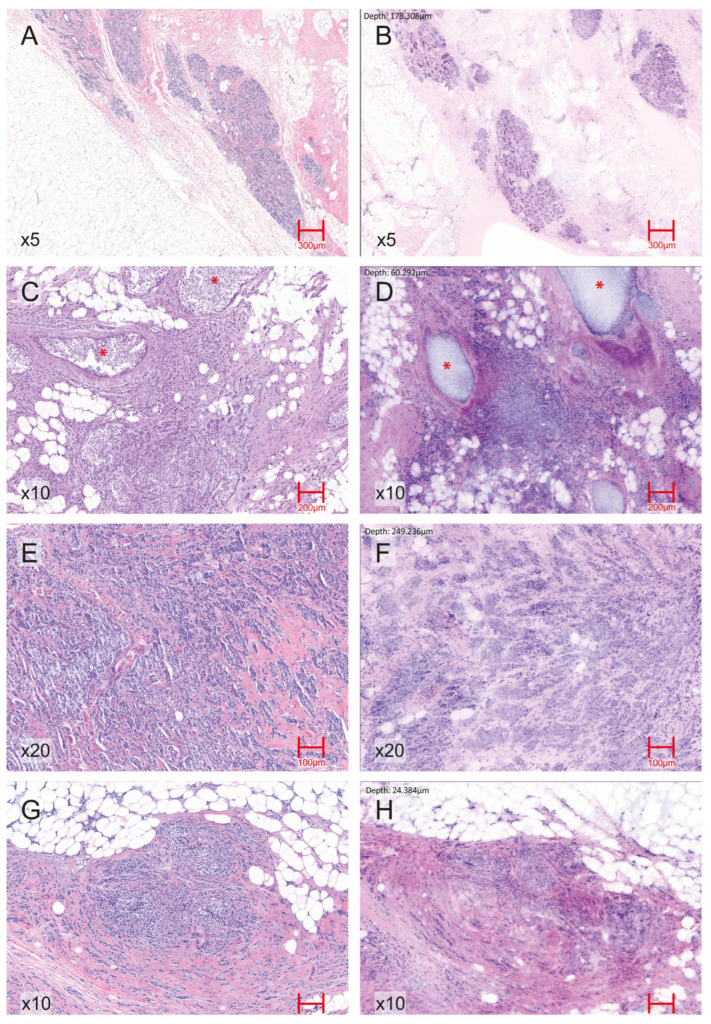
Representative EVFCM–H&E correlation across normal breast tissue and key histopathological breast carcinoma patterns. (**A**) H&E image (5× magnification) of normal breast parenchyma showing terminal duct–lobular units and interlobular stroma. (**B**) Corresponding EVFCM image (5× magnification) of normal breast tissue reproducing the lobular architecture and stromal features. (**C**) H&E section (10× magnification) of IDC with associated ductal carcinoma in situ (DCIS) component, highlighted with red asterisks (*). (**D**) Corresponding EVFCM image (10× magnification) demonstrating the same in situ foci (*) adjacent to invasive tumour nests. (**E**) H&E image (20× magnification) of IDC showing infiltrative malignant epithelial clusters with desmoplastic stromal reaction. (**F**) EVFCM image of the same region (20× magnification), illustrating similar architectural and cytological features, including cohesive tumour nests and stromal changes. (**G**) H&E section (10× magnification) of invasive lobular carcinoma displaying discohesive small cells arranged in single files and linear cords infiltrating the stroma. (**H**) Corresponding EVFCM image (10× magnification), identifying the same lobular growth pattern and cell morphology.

**Figure 3 cancers-17-03848-f003:**
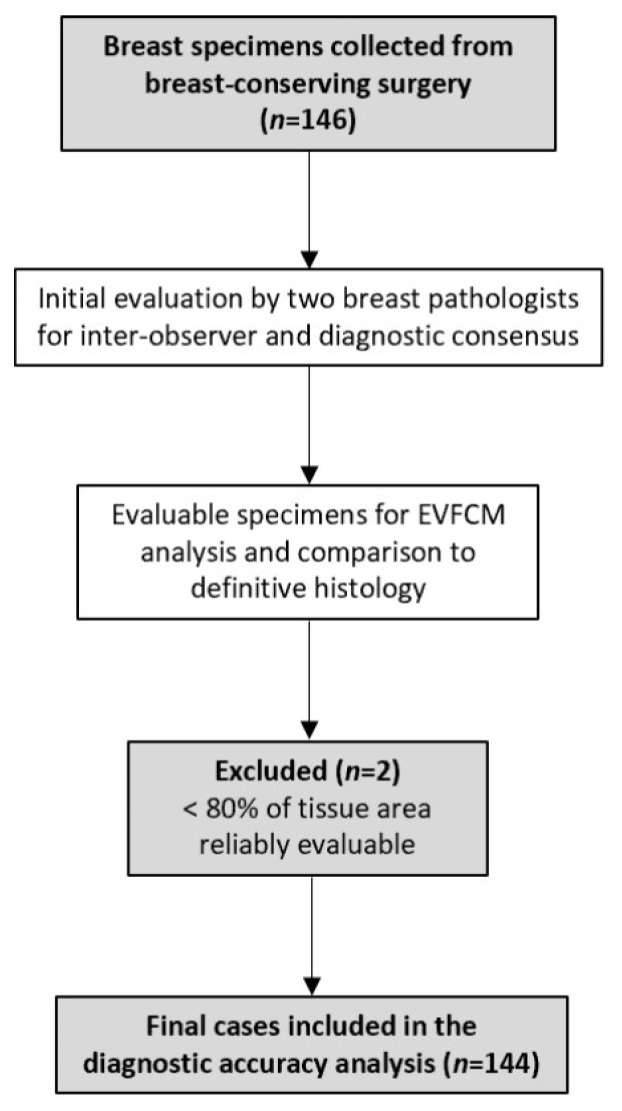
STARD flow diagram illustrating sample inclusion and final cases analysed.

**Table 1 cancers-17-03848-t001:** Inter-observer agreement and distribution of EVFCM readings between pathologists.

Variable	Category	Pathologist A	Pathologist B	Agreement %	Cohen’s κ (95% CI)	Interpretation
Neoplasia (*n* = 146)	No	88 (60.3%)	90 (61.6%)	97.3	0.942 (0.881–0.986)	Almost perfect
	Yes	58 (39.7%)	56 (38.4%)			
Tumour type (*n* = 146)	No	88 (60.3%)	90 (61.6%)	93.8	0.883 (0.807–0.948)	Almost perfect
	Invasive	49 (33.6%)	42 (28.8%)			
	In situ	8 (5.5%)	8 (5.5%)			
	Invasive + In situ	1 (0.7%)	6 (4.1%)			
Invasive subtype (*n* = 146)	No	88 (60.3%)	90 (61.6%)	97.3	0.946 (0.885–0.987)	Almost perfect
	Ductal	52 (35.6%)	50 (34.2%)			
	Lobular	3 (2.1%)	3 (2.1%)			
	Other	3 (2.1%)	3 (2.1%)			
Artefact (*n* =146)	No	131 (89.7%)	120 (82.2%)	84.2	0.355 (0.143–0.547)	Fair
	Yes	15 (10.3%)	26 (17.8%)			
Suitability (*n* = 146)	Suitable	144 (98.6%)	142 (97.3%)	98.6	0.651 (0.535–0.767)	Substantial
	Not suitable	2 (1.4%)	4 (2.7%)			

Percentage of agreement and Cohen’s κ values (95% CI) for the inter-observer evaluation of EVFCM parameters. Agreement strength was interpreted according to Landis and Koch’s criteria.

**Table 2 cancers-17-03848-t002:** Comparison of EVFCM consensus with paraffin histology for diagnostic performance.

Variable	Category	Sensitivity %	Specificity %	PPV %	NPV %	Overall Accuracy %	Cohen’s k	*p*-Value
Neoplasia (*n* = 144)	Yes (62) No (82)	93.7 98.8	98.8 93.7	98.3 95.3	95.3 98.3	96.6	0.929 (0.861–0.986)	<0.001
Tumour type (*n* = 144)						95.8	0.925 (0.859–0.975)	<0.001
	No (82)	98.8	93.5	95.3	98.3			
	Invasive (47)	91.5	99	97.7	96			
	In situ (10)	90	99.3	90	99.3			
	Invasive + In situ (5)	100	100	100	100			
Invasive subtype (*n* = 144)						95.1	0.907 (0.831–0.970)	<0.001
	No (82)	98.8	93.7	95.3	98.3			
	Ductal (54) Lobular (5) Other (3)	96.3 40 66.7	97.8 100 99.3	96.3 100 66.7	97.8 97.9 99.3			
Margin (*n* = 144)						99.3	0.857 (0.5–1)	<0.001
	Positive (5) Negative (139)	80 100	100 80	100 99.3	99.3 100			
Distance to margin (*n* = 144)						99.3	0.854 (0.566–1)	<0.001
	<2 mm (4) ≥2 mm (140)	75 100	100 75	100 95.7	99.3 100			

Sensitivity, specificity, positive predictive value (PPV), negative predictive value (NPV), and Cohen’s κ for each diagnostic category were calculated. Two samples were excluded from the analysis due to artefacts, as determined by consensus among the pathologists. Paraffin-embedded histology served as the gold standard, and the sample distribution (*n*) corresponds to paraffin diagnosis.

## Data Availability

The data supporting the findings of this study are available from the corresponding author upon reasonable request. Access will be granted following a formal request and completion of the required data-sharing agreements, in compliance with ethical and privacy regulations.

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
