# Peer review of "Towards a New Standard: Prospective Validation of Ex Vivo Fusion Confocal Microscopy for Intraoperative Margin Assessment in Breast-Conserving Cancer Surgery"

_cancers, 2025, doi:10.3390/cancers17233848_

Round 1
Reviewer 1 Report
Comments and Suggestions for Authors
I have read the article "Towards a New Standard: Prospective Validation of Ex Vivo Fusion Confocal Microscopy for Intraoperative Margin Assessment in Breast-Conserving Surgery".
The authors thoroughly describe their experience with 144 breast tissue specimens examined intraoperatively.
They have demonstrated high concordance between two observers in comparing confocal microscopy and frozen-section analysis.
It would be of interest to compare the overall macroscopic impression of the margin's status with the microscopic assessment because macroscopic selection of tissue is of great importance for the correct evaluation of margins.
Importantly, the intraoperative assessment time is reasonable—around 5 minutes—and does not significantly prolong surgical time.
As the authors indicate, further experience should be gained with a larger number of cases, especially those including ductal carcinoma in situ, multifocality, and close margins (<2 mm), which are known to be problematic.
Overall, the article is well written and illustrated.
The references are complete and up to date.
I think this article would be of great interest to specialists involved in assessing the margins of resection in breast surgical specimens.
Author Response
Comments and Suggestions for Authors
I have read the article "Towards a New Standard: Prospective Validation of Ex Vivo Fusion Confocal Microscopy for Intraoperative Margin Assessment in Breast-Conserving Surgery".
The authors thoroughly describe their experience with 144 breast tissue specimens examined intraoperatively. They have demonstrated high concordance between two observers in comparing confocal microscopy and frozen-section analysis. It would be of interest to compare the overall macroscopic impression of the margin's status with the microscopic assessment because macroscopic selection of tissue is of great importance for the correct evaluation of margins. Importantly, the intraoperative assessment time is reasonable—around 5 minutes—and does not significantly prolong surgical time.
As the authors indicate, further experience should be gained with a larger number of cases, especially those including ductal carcinoma in situ, multifocality, and close margins (<2 mm), which are known to be problematic. Overall, the article is well written and illustrated. The references are complete and up to date. I think this article would be of great interest to specialists involved in assessing the margins of resection in breast surgical specimens.
Response:
We sincerely thank the reviewer for their thoughtful and positive evaluation of our manuscript, and for acknowledging its clarity, comprehensiveness, and clinical relevance. We particularly appreciate the insightful comment regarding the correlation between the macroscopic impression of the margins and their microscopic assessment. We fully agree that macroscopic selection of the area to be analysed is a critical determinant for accurate intraoperative margin evaluation, as it directly affects the representativeness and reliability of the microscopic findings.
In our current workflow, macroscopic orientation and targeted selection were performed collaboratively by the breast surgeon and the pathologist immediately after excision. This joint approach ensured that the regions judged most suspicious for residual disease were prioritised for EVFCM imaging, effectively aligning macroscopic judgement with subsequent microscopic evaluation. To clarify this integration, we have expanded the description in the Materials and Methods (Section 2.2, Surgical Procedure) to explicitly highlight the coordinated contribution between the surgical and pathological teams during processing.
We also fully concur with the reviewer’s valuable suggestion regarding the importance of expanding future experience, particularly in challenging scenarios such as ductal carcinoma in situ, multifocal disease, and narrow margins (< 2 mm). This point has now been incorporated into the Discussion, emphasising that future multicentric studies should specifically address these subgroups to confirm reproducibility, optimise interpretative criteria, and strengthen the generalisability of EVFCM in complex histopathological contexts.
We are grateful for the reviewer’s supportive comments and constructive suggestions, which have helped to further enhance the clarity and completeness of the manuscript. These modifications have been introduced in the revised version of the manuscript, with all changes marked in red to ensure their straightforward and rapid identification by the reviewers.
Changes in manuscript:
- Section: 2.2 Surgical Procedure (Lines 158 - 162)
Revised text added:
“Macroscopic inspection and orientation were jointly performed by the breast surgeon and the pathologist immediately after excision. This collaboration ensured that the areas most suspicious for residual disease were selected for EVFCM imaging, aligning macroscopic judgement with microscopic evaluation. In our workflow, EVFCM assessment was integrated at this same decision point, functionally replacing frozen section analysis. - Section: 4. Discussion (Lines 553 - 559)
Added clarification:
“In addition, further multicentric research should specifically address cases known to pose particular diagnostic challenges, including ductal carcinoma in situ (DCIS), multifocal disease, and narrow margins (<2 mm). These scenarios often present complex or subtle morphological patterns that can hinder intraoperative recognition, even in conventional histopathology. Expanding the validation of EVFCM across these sub-groups will be essential to confirm its reproducibility, refine interpretative criteria, and ensure its applicability in all clinically relevant contexts of breast-conserving surgery.”
Reviewer 2 Report
Comments and Suggestions for Authors
The authors present an interesting research project using the VivaScope 2500M-G4 for margin assessment in Ex Vivo stained fresh breast tissue specimens. Overall the work is important; digital pathology has the potential to improve patient outcomes and aid pathologists, especially in times of disagreement or edge cases. However, I believe the work should be improved prior to publication. All of my suggested revisions are achievable without any subsequent experiments. Please see my suggestions below:
Major
- Section 2.3 "Staining Protocol, Image Acquisition and Histopathological Processing" is missing many key details. If a researcher were trying to replicate the findings, the would need much more information:
- Objective magnification and numerical aperture. Preferably also objective type if it has any spherical or chromatic aberration correction.
- Laser power, gain settings, pixel dwell times, pixel/voxel sizes.
- Information about the stitching and tilescanning. The authors claim the image is acquired in less than 5 minutes. The field-of-view size, % overlap between adjacent tiles, stitching process (I assume done by the microscope's software), are all important.
- The authors state "optical sectioning of fresh tissue was obtained up to a depth of approximately 200µm, with optimal resolution within the superficial 150µm layer of tissue". This could be worded clearer. The term optical sectioning suggests z-stacking. But it is unclear whether a volumetric image is acquired. Nothing in the processing or results suggests a z-stack was used for analysis. Are you taking a z-stack? If so, what is the slice interval. My assumption is that a full volumetric image of a 3x3 cm specimen using the 38 x magnification, 0.85 NA water immersion (from the Vivascope technical datasheet) would take much longer than 5 minutes. The term "optimal resolution" suggests the software has a setting to set the pixel and slice interval to satisfy Nyquist sampling. Is this what you mean? If so, please state explicitly.
- Related to my last point, if optical sectioning/z-stacking was performed, was the analysis done on 2D images like the ones presented in figures 1&2? Was a maximum intensity projection done on a volumetric image to get the resulting 2D image? Or was a single plane chosen near the surface?
- It may seem obvious, but the emission window of Acridine Orange should also be reported.
- FGFCF has an excitation peak of ~625nm and emission of ~640nm. The 638nm channel is referred to as reflectance only and I am aware that the VivaScope often refers to this channel as such, but this suggests it is also being used to some extent with the FGFCF stain, in which case it would seem it is also fluorescence.
- "Images were displayed using proprietary software" is not sufficient. If it is the onboard algorithm that VivaScope uses for pseudocoloring, just state that.
- What is the bit-depth of the TIFF format image?
- "Following EVFCM imaging"... do the stains/dyes need to be washed out prior to the histopathological prep?
- "Staining with H&E to establish a definitive diagnosis"... the image acquisition parameters for imaging the H&E slides should also be reported.
- Some of these details are available on the VivaScope technical datasheet (like the objective), which is cited, but it is not appropriate to make the reader hunt for the details and we should endeavor to report as completely as possible in the spirit of open science and repeatability.
- Some of the conclusions and claims are not fully supported by the data.
- Line 364 "Four of the five... demonstrating the strong diagnostic performance of EVFCM". A data subset of 80% of 5 samples is not sufficient to claim this is strong diagnostic performance.
- Line 350 "A lower sensitivity (40%) was...". Again, this data subset for lobular or other invasive subtypes is too limited to draw definitive conclusions.
- In line 82 of the introduction, the authors point out that reoperation rates exceed 20% (is this worldwide?). Given that the positive margin sensitivity of the study is 80%, this suggests the instrument would be no better, leading to 20% of specimens with positive margins being missed, potentially leading to a necessary subsequent operation.
- I strongly recommend inserting the word preliminary in claims where your replicate number is less than 5. This limitation is discussed in lines 428-435, but that does not make up for the fact that some of the claims prior are not definitively supported due to a small unbalanced dataset. It is certainly good that the surgical team is so adept at removing the whole tumor, though.
Minor:
- The title uses the word "intraoperative". It seems the work has the potential to be used intraoperatively, but it does not look like this work was performed intraoperatively. "Following resection, specimens and margins were delivered fresh to the Department of Pathology". Is the Vivascope and Pathology Department actually close enough to the operating suite to claim this?
- The abstract says 144 specimens, but the materials and methods says 146. I know from figure 3 and the caption of table 2 why 2 were excluded, but this exclusion should be stated in line 214-215 where the exclusion criteria is presented.
- The depth text overlay in figures 1 and 2 is too small to read easily. In some cases the scale bar text is also difficult to read due to the color contrast of the underlying image.
- Table 1, artefact agreement. Why is there so much disagreement in what constitutes an artefact between the pathologists? This should be addressed somewhere in text. In some ways (in my opinion) this disagreement actually strengthens the need for digital pathology methods like this research.
- Conclusions, line 479. "workflow completed in under five minutes". If image acquisition takes "less than 5 minutes per sample" (line 193), and the sample needs to first be treated an stained for ~1 minute, and the tissue needed to be moved to the pathology department, handled and oriented, and then the image needs to be processed by the VivaScope algorithms, and finally it needs to be interpreted, then I highly doubt the timeline works out to less than 5 minutes. The workflow certainly seems rapid, but this <5-minute total claim seems inaccurate given the description of steps in the methods.
Author Response
Comments and Suggestions for Authors
The authors present an interesting research project using the VivaScope 2500M-G4 for margin assessment in Ex Vivo stained fresh breast tissue specimens. Overall the work is important; digital pathology has the potential to improve patient outcomes and aid pathologists, especially in times of disagreement or edge cases. However, I believe the work should be improved prior to publication. All of my suggested revisions are achievable without any subsequent experiments. Please see my suggestions below:
We sincerely thank Reviewer for their thoughtful and constructive evaluation of our manuscript. We greatly appreciate the acknowledgement of the scientific relevance of our work and of the potential of digital pathology to enhance intraoperative decision-making. We have carefully addressed each of the reviewer’s comments in full detail, and all requested clarifications and methodological specifications have now been incorporated into the revised version of the manuscript.
To facilitate a clear and efficient review, all modifications have been highlighted in red, enabling straightforward identification of the changes introduced. We believe these revisions have significantly strengthened the transparency, reproducibility, and overall clarity of the study. A point-by-point response is provided below.
Major
1. Section 2.3 "Staining Protocol, Image Acquisition and Histopathological Processing" is missing many key details. If a researcher were trying to replicate the findings, the would need much more information:
- Objective magnification and numerical aperture. Preferably also objective type if it has any spherical or chromatic aberration correction.
Response:
We thank the reviewer for this precise and constructive observation. To enhance the methodological transparency and reproducibility of our study, we have now included the complete optical specifications of the VivaScope 2500M-G4 objective in Section 2.3. According to the manufacturer’s technical documentation (VivaScope 2500M-G4 Technical Datasheet, MAVIG GmbH, Munich, Germany), the system is equipped with a 38×, 0.85 numerical aperture (NA) water-immersion objective that incorporates spherical and chromatic aberration correction. This configuration is specifically optimised for high-resolution en face confocal imaging of fresh, unfixed tissue, providing a lateral optical resolution of <1.25 µm and an axial resolution of <5 µm at the centre of the field of view. These details have been added to the revised manuscript to ensure that readers can accurately replicate the imaging configuration with comparable systems.
Changes in manuscript:
- Section: 2.3. Staining Protocol, Image Acquisition and Histopathological Processing (Lines 212-216)
Added text:
“This device is equipped with a 38×, 0.85 numerical aperture (NA) water-immersion objective with spherical and chromatic aberration correction optimised for en-face confocal imaging of fresh tissue. This optical configuration provides a lateral optical resolution of <1.25 µm and an axial resolution of <5 µm at the centre of the field of view.”
- Laser power, gain settings, pixel dwell times, pixel/voxel sizes.
Response
We thank the reviewer for highlighting the importance of clearly reporting acquisition parameters to ensure methodological reproducibility. The VivaScope 2500M-G4 operates as a closed, fully automated confocal imaging system that uses factory-calibrated acquisition presets. All images in this study were acquired using the manufacturer’s standard preset for fresh ex vivo tissue, which automatically regulates laser power, detector gain, pixel dwell time, and sampling density according to predefined algorithms optimised for tissue contrast and resolution.
Because these parameters are not user-adjustable and are internally managed by the proprietary software, they cannot be reported numerically; however, all scans were obtained under identical preset conditions, ensuring high inter-sample reproducibility. The system’s pixel sampling corresponds to the native optical resolution of the 38×/0.85 NA (numerical aperture) objective and satisfies Nyquist sampling criteria for accurate reconstruction of nuclear and stromal detail. This clarification has been added to the revised text in Section 2.3.
Changes in manuscript:
- Section: 2.3. Staining Protocol, Image Acquisition and Histopathological Processing (Lines 220-226)
Added text:
“All EVFCM acquisitions were performed using the manufacturer’s preset for fresh ex vivo tissue, which automatically adjusts laser power, detector gain, pixel dwell time, and sampling density to optimise image contrast and resolution. These parameters are factory-calibrated, not user-modifiable, and ensure consistency and reproducibility across samples. Pixel sampling corresponds to the system’s native optical resolution and complies with Nyquist sampling criteria, providing sufficient detail for accurate assessment of nuclear and architectural morphology.”
- Information about the stitching and tilescanning. The authors claim the image is acquired in less than 5 minutes. The field-of-view size, % overlap between adjacent tiles, stitching process (I assume done by the microscope's software), are all important.
Response:
We thank the reviewer for this detailed and constructive comment. The VivaScope 2500M-G4 includes an automated mosaic-acquisition function that captures adjacent confocal fields of view and merges them into a continuous en-face image through the system’s proprietary software. The alignment and blending of tiles are performed internally by the system, and no manual image registration, stitching adjustments, or post-processing were required in this study.
The manufacturer does not publicly disclose the numerical parameters related to the field-of-view dimensions per tile or the percentage of overlap between adjacent images. These parameters are factory-calibrated and verified during the system’s quality-control process, ensuring accurate geometric alignment and reproducible image reconstruction across all acquisitions. Consequently, all samples in our study were imaged under identical conditions using the same automated preset, guaranteeing methodological consistency and reproducibility across the cohort.
In our workflow, the total area scanned corresponded to the portion of the specimen surface selected for evaluation, typically representing the region most at risk for residual disease based on macroscopic inspection. The complete EVFCM process—including tissue staining, positioning, and image acquisition—was consistently completed in less than five minutes per specimen, in line with the manufacturer’s operational specifications and our practical experience.
To improve clarity, we have added information to Section 2.3 of the revised manuscript explaining that mosaic generation and tile stitching were performed automatically by the system’s software.
Changes in manuscript:
- Section: 3. Staining Protocol, Image Acquisition and Histopathological Processing (Lines 236-242)
Added text:
“Large-area confocal mosaics were obtained using the system’s automated mosaic-acquisition function, which captures adjacent fields of view and merges them through the system’s proprietary software to produce a continuous en-face image. The manufacturer does not disclose tile-overlap percentages or field-of-view dimensions for individual tiles; these values are internally calibrated and verified during quality-control procedures to ensure alignment accuracy. All specimens were imaged under identical automated settings.” - The authors state "optical sectioning of fresh tissue was obtained up to a depth of approximately 200µm, with optimal resolution within the superficial 150µm layer of tissue". This could be worded clearer. The term optical sectioning suggests z-stacking. But it is unclear whether a volumetric image is acquired. Nothing in the processing or results suggests a z-stack was used for analysis. Are you taking a z-stack? If so, what is the slice interval. My assumption is that a full volumetric image of a 3x3 cm specimen using the 38 x magnification, 0.85 NA water immersion (from the Vivascope technical datasheet) would take much longer than 5 minutes. The term "optimal resolution" suggests the software has a setting to set the pixel and slice interval to satisfy Nyquist sampling. Is this what you mean? If so, please state explicitly.
Response:
We thank the reviewer for this valuable observation. The term “optical sectioning” in our manuscript refers to the intrinsic property of confocal microscopy to obtain thin, in-focus optical slices without the need for physical sectioning of the specimen. In this study, the VivaScope 2500M-G4 was operated in en-face mode, producing two-dimensional confocal images from the superficial layer of fresh breast tissue.
No z-stack or volumetric image reconstruction was performed. Each image corresponds to a single optical plane obtained at a fixed focal depth. The system allows focusing at variable depths within the tissue up to approximately 200 µm, although optimal image resolution and signal-to-noise ratio are achieved within the upper 100–150 µm, as indicated by the manufacturer.
The expression “optimal resolution” refers to this empirically defined imaging depth and not to a user-adjustable Nyquist sampling setting. The VivaScope 2500M-G4 operates with preset optical and pixel parameters that are automatically calibrated by the system’s proprietary software to ensure consistent imaging performance.
These clarifications have been incorporated into Section 2.3 of the revised manuscript to enhance terminological precision and methodological transparency.
Changes in manuscript:
Section: 2.3. Staining Protocol, Image Acquisition and Histopathological Processing
Added text:
(Lines 227-231):
“The system enables optical focusing of fresh tissue up to a depth of approximately 200 μm, with an optimal image resolution and signal-to-noise ratio achieved within the superficial 100-150 μm layer of the tissue. This depth range is suitable for the intraoperative assessment of surface margins, enabling the detection of tumour cells at or near the resection interface.”
(Lines 231-232):
“Confocal imaging was performed in en-face mode, generating two-dimensional optical sections from the superficial tissue layer as a single optical plane.”
(Lines 234-235):
“No volumetric (z-stack) acquisition, projection methods, or post-acquisition processing were applied.”
(Lines 248-251):
“The EVFCM processing phase defined as staining, mounting and image acquisition once the specimen had been received and oriented in the pathology laboratory was consistently completed in under 5 minutes per sample.”
- Related to my last point, if optical sectioning/z-stacking was performed, was the analysis done on 2D images like the ones presented in figures 1&2? Was a maximum intensity projection done on a volumetric image to get the resulting 2D image? Or was a single plane chosen near the surface?
Response:
We thank the reviewer for this follow-up clarification. As detailed above, no z-stack or volumetric image acquisition was performed in this study. All analyses were conducted on two-dimensional (en-face) confocal images acquired directly from the tissue surface.
Each image corresponds to a single optical plane obtained at the depth providing the best cellular contrast and fluorescence signal, typically within the upper 100–150 µm of the specimen. The VivaScope 2500M-G4 automatically determines this focal plane during acquisition based on real-time optimisation of laser focus and signal intensity.
No maximum intensity projection or volumetric rendering was applied. The images presented in Figures 1 and 2 reflect the raw output of the system following real-time pseudocoloured conversion by the proprietary software, without additional post-processing.
This clarification has been added to Section 2.3 of the revised manuscript to specify that analysis was performed on single-plane 2D confocal images.
Changes in manuscript:
- Section: 2.3. Staining Protocol, Image Acquisition and Histopathological Processing
Added text:
(Lines 231-232):
“Confocal imaging was performed in en-face mode, generating two-dimensional optical sections from the superficial tissue layer as a single optical plane.”
(Lines 234-235):
“No volumetric (z-stack) acquisition, projection methods, or post-acquisition processing were applied.”
- It may seem obvious, but the emission window of Acridine Orange should also be reported.
Response:
We thank the reviewer for this helpful observation. Acridine Orange (AO) was used as a fluorescent nuclear stain to enhance contrast between epithelial and stromal components. In our protocol, AO fluorescence was excited at 488 nm, and emission was collected at wavelengths above 500 nm, consistent with the optical configuration of the VivaScope 2500M-G4 system.
This clarification has been incorporated into Section 2.3 of the revised manuscript to ensure full methodological transparency.
Changes in manuscript:
- Section: 2.3. Staining Protocol, Image Acquisition and Histopathological Processing (Lines 203-206)
Added text:
“Acridine orange is a fluorescent dye that binds specifically to nucleic acids and emits green fluorescence upon excitation at 488 nm, enabling clear visualisation of nu-clear structures in confocal images [10]. The emitted fluorescence of acridine orange was collected above 500 nm.”
- FGFCF has an excitation peak of ~625nm and emission of ~640nm. The 638nm channel is referred to as reflectance only and I am aware that the VivaScope often refers to this channel as such, but this suggests it is also being used to some extent with the FGFCF stain, in which case it would seem it is also fluorescence.
Response:
We appreciate the reviewer’s precise and insightful comment. In the VivaScope 2500M-G4, the 638 nm channel operates strictly in reflectance mode, as specified by the manufacturer. Fast Green FCF (FGFCF) was applied solely as a counterstain to enhance cytoplasmic and stromal contrast through light absorption, not as a fluorescent marker. Although the excitation and emission peaks of FGFCF (~625 nm and ~640 nm) are close to the wavelength of the reflectance laser, the system does not excite or detect fluorescence from this dye.
The signal acquired at 638 nm corresponds exclusively to reflected light from the specimen surface, which is then merged with the 488 nm fluorescence channel (Acridine Orange) by the proprietary imaging software to generate the final pseudocoloured composite image.
This clarification has been incorporated into Section 2.3 of the revised manuscript to explicitly distinguish between fluorescence and reflectance signal acquisition.
Changes in manuscript:
- Section: 2.3. Staining Protocol, Image Acquisition and Histopathological Processing
Added text:
(Lines 206-208):
“Fast green FCF stains the cytoplasm and extracellular matrix, enhancing the structural details, including fibrotic areas [39], contributing only to reflectance-based contrast and was not excited as a fluorescent dye.
(Lines 216-219):
“This system employs two simultaneous laser channels—fluorescence (488 nm) with excitation of acridine orange and reflectance (638 nm)—which are automatically merged by the proprietary acquisition software to produce high-resolution fused confocal images”.
- Images were displayed using proprietary software" is not sufficient. If it is the onboard algorithm that VivaScope uses for pseudocoloring, just state that.
Response:
We thank the reviewer for this helpful clarification. Images were visualised using the VivaScope onboard proprietary software, which automatically generates the pseudocoloured composite image by combining the 488 nm fluorescence channel with the 638 nm reflectance channel through its integrated rendering algorithm. No external software or additional post-processing was applied.
This has been specified in Section 2.3 of the revised manuscript.
Changes in manuscript:
- Section: 2.3. Staining Protocol, Image Acquisition and Histopathological Processing (Lines 243-247)
Added text:
“The images were visualised using the VivaScope onboard proprietary software, which automatically applies the system’s integrated pseudocoloring algorithm to merge the 488 nm fluorescence and 638 nm reflectance channels into a single composite image that simulates H&E staining. The images analysed represent the direct output generated by the system, with no external software or additional post-processing applied.”
- What is the bit-depth of the TIFF format image?
Response:
We thank the reviewer for this question. All images acquired with the VivaScope 2500M-G4 were exported in 8-bit TIFF format (uint8) as single-plane files, using the instrument’s proprietary software. These pseudocoloured composite images are generated automatically by combining the 488 nm fluorescence and 638 nm reflectance signals through the onboard algorithm. No external processing or bit-depth conversion was applied.
This specification has been added to Section 2.3 of the revised manuscript.
Changes in manuscript:
- Section: 3. Staining Protocol, Image Acquisition and Histopathological Processing (Lines 252-255)
Added text:
“Images were stored and exported in 8-bit TIFF format while preserving the full spatial resolution of the acquisition as single-plane files with pseudocoloring automat-ically generated by the device software, and no external conversion or bit-depth mod-ification was performed to ensure traceability and support a structured diagnostic re-view.”
- "Following EVFCM imaging"... do the stains/dyes need to be washed out prior to the histopathological prep?
Response:
We thank the reviewer for this important clarification. No additional washing or destaining step was required after EVFCM imaging. Both Acridine Orange and Fast Green FCF are applied superficially and show minimal penetration into the deeper tissue layers, acting primarily as surface contrast agents for rapid confocal acquisition.
During routine formalin fixation and standard histopathological processing and paraffin embedding both dyes are fully removed. In our series, no residual staining or interference with H&E morphology was observed in any paraffin-embedded section.
For this reason, specimens proceeded directly from EVFCM imaging to histopathological preparation without requiring any intermediate washing step. This clarification has been incorporated into Section 2.3 of the revised manuscript.
Changes in manuscript:
- Section: 2.3. Staining Protocol, Image Acquisition and Histopathological Processing (Lines 256 - 261)
Added text:
“Following EVFCM imaging, all samples underwent routine formalin fixation, paraffin embedding, sectioning, and staining with H&E to establish a definitive diagnosis. No washing or destaining step was required. Acridine Orange and Fast Green FCF are applied superficially and exhibit minimal tissue penetration. Both dyes are removed during routine fixation, dehydration, and paraffin embedding, without affecting subsequent H&E staining or histopathological interpretation.”
- "Staining with H&E to establish a definitive diagnosis"... the image acquisition parameters for imaging the H&E slides should also be reported.
Response:
We thank the reviewer for this valuable observation. All H&E-stained slides were digitized using the routine whole-slide scanner of our Pathology Department (PANNORAMIC 100 flash DX (3DHISTECH Lts., Budapest, Hungary)(P1000). Slides were scanned at 20× magnification (0.24 μm/pixel resolution) and images were exported in MRXS format and reviewed using ClinicalViewer (3DHISTECH Ltd., Budapest, Hungary). No additional post-processing was performed.
This information has been added to Section 2.3 of the revised manuscript.
Changes in manuscript:
- Section: 3. Staining Protocol, Image Acquisition and Histopathological Processing (Lines 264 - 267)
Added text:
“H&E-stained slides were digitised using the departmental whole-slide scanner PANNORAMIC 100 flash DX (3DHISTECH Lts., Budapest, Hungary)(P1000) at 20× magnification (0.24 μm/pixel), exported in MRXS format, and reviewed using ClinicalViewer (3DHISTECH Ltd., Budapest, Hungary). No additional post-processing was performed.”
- Some of these details are available on the VivaScope technical datasheet (like the objective), which is cited, but it is not appropriate to make the reader hunt for the details and we should endeavor to report as completely as possible in the spirit of open science and repeatability.
Response:
We fully agree with the reviewer’s observation. In response to this helpful comment, we have revised Section 2.3 extensively to ensure that all essential technical details are explicitly reported within the manuscript itself, without requiring readers to consult the external VivaScope technical datasheet.
Following the reviewer’s constructive guidance, we now include the full optical specifications of the objective (magnification, numerical aperture, aberration correction), the excitation and emission characteristics of the dyes, the operating wavelengths of both laser channels, the acquisition principles (including the absence of z-stacking), the automated mosaic-generation process, the pseudocolouring algorithm applied by the onboard software, and the bit-depth and format of exported TIFF images. We have also added details regarding the handling of dyes prior to histopathological processing and the digitisation parameters for H&E-stained slides.
Thanks to the reviewer’s precise and thoughtful suggestions, Section 2.3 has been substantially strengthened. These additions significantly enhance methodological transparency, improve reproducibility, and align the manuscript with the principles of open science.
2. Some of the conclusions and claims are not fully supported by the data.
- Line 364 "Four of the five... demonstrating the strong diagnostic performance of EVFCM". A data subset of 80% of 5 samples is not sufficient to claim this is strong diagnostic performance.
Response:
We thank the reviewer for this important remark and fully agree that the small number of positive-margin cases (n=5) limits the robustness of the sensitivity estimate. In the original version, the phrase “demonstrating the strong diagnostic performance of EVFCM” was indeed too emphatic given the small subset of involved margins.
In the revised manuscript, we have therefore tempered this statement to better reflect the exploratory nature of these data. We now report that EVFCM correctly identified 4 out of 5 histologically involved margins and explicitly acknowledge that, although these findings are encouraging, the sensitivity estimate should be interpreted with caution and confirmed in larger cohorts.
This change aligns our wording with the sample size and with the limitations already discussed in the manuscript.
Changes in manuscript:
- Section: Discussion (Lines 445-451)
“In this study, the EVFCM achieved a sensitivity of 80%, specificity of 100%, and an overall diagnostic accuracy of 99.3% for margin detection (κ=0.857, p<0.001). Four of the five histologically involved margin cases were correctly identified using EVFCM. Although the number of positive margin cases is small, these preliminary findings are encouraging and support the potential of EVFCM for intraoperative margin assess-ment, warranting confirmation in larger series”
- Line 350 "A lower sensitivity (40%) was...". Again, this data subset for lobular or other invasive subtypes is too limited to draw definitive conclusions.
Response:
We sincerely thank the reviewer for this highly accurate and important observation. We fully agree that the number of invasive lobular carcinoma (ILC) cases in our cohort was too small to support any definitive conclusions regarding subtype-specific diagnostic performance. The reported sensitivity of 40% reflects only a very limited subset, and as such it carries substantial statistical uncertainty.
Our intention in the original manuscript was to acknowledge the well-known diagnostic challenges associated with ILC—challenges observed not only in confocal imaging, but also in conventional histopathology due to its discohesive growth pattern and subtle architectural distortion. However, we recognise that the wording may have inadvertently implied a level of confidence that is not justified by the small sample size.
In response to the reviewer’s comment, we have revised this section to more clearly state that the observed sensitivity should be regarded as preliminary, not conclusive, and that no firm claims can be made regarding ILC-specific performance based on our dataset alone. We also emphasise the need for larger, subtype-balanced cohorts in future validation studies to reliably determine EVFCM performance across different invasive subtypes.
We thank the reviewer for prompting this clarification, which has improved the precision and interpretability of the manuscript.
Changes in manuscript:
- Section: Discussion (Lines 439-443)
“However, given the small number of ILC cases included in our cohort, this sensitivity value should be interpreted as preliminary rather than conclusive. Larger and more balanced datasets are required to establish robust subtype-specific performance metrics and to validate whether this trend persists in broader clinical settings.”
- In line 82 of the introduction, the authors point out that reoperation rates exceed 20% (is this worldwide?). Given that the positive margin sensitivity of the study is 80%, this suggests the instrument would be no better, leading to 20% of specimens with positive margins being missed, potentially leading to a necessary subsequent operation.
Response:
We sincerely thank the reviewer for this important and thoughtful comment. In the revised manuscript, we now clarify that the >20% reoperation rate cited in line 82 refers to figures reported across multiple international series and meta-analyses of breast-conserving surgery, rather than reflecting a single healthcare system.
We fully agree that this clinical reoperation rate cannot be directly compared with the 80% sensitivity for margin positivity observed in our study. Our investigation was specifically designed as a prospective diagnostic validation of EVFCM relative to paraffin-embedded histology, and not to evaluate the impact of EVFCM on re-excision rates. As the reviewer correctly notes, reoperation rates arise from a complex interplay of factors—including tumour biology and multifocality, surgical technique, margin definitions, use of cavity shavings, pathological workflow, and institutional practice patterns—many of which extend well beyond the diagnostic performance of any intraoperative imaging method.
Importantly, the sensitivity estimate in our cohort is based on only five truly involved margins and is therefore inherently imprecise. A focused review of these cases showed that the single margin not detected by EVFCM corresponded to ductal carcinoma in situ (DCIS) associated with epithelial hyperplasia at the resection edge—a scenario characterised by substantial morphological overlap and widely recognised as one of the most challenging situations in both frozen-section evaluation and optical-based intraoperative techniques. As such, the calculated sensitivity is highly sensitive to single-case variation and should be interpreted as preliminary.
When considering the entire cohort of 144 evaluable specimens, the overall diagnostic accuracy for margin assessment was 99.3%, with almost perfect interobserver agreement (κ = 0.857). These findings support the robustness of EVFCM in this setting, while underscoring that larger, multicentric studies with greater numbers of margin-positive cases will be required to establish stable and generalisable sensitivity estimates.
To avoid any unintended extrapolation between diagnostic sensitivity and clinical reoperation rates, both the Introduction and Discussion have been revised to state explicitly that this study did not assess the effect of EVFCM on reoperation outcomes, and that outcome-focused investigations will be necessary to determine whether the integration of EVFCM can contribute to reducing re-excision rates.
We are grateful to the reviewer for raising this point, which has prompted meaningful clarifications and strengthened the overall contextual precision of the manuscript.
Changes in manuscript:
- Section: 1. Introduction (Lines 82-84)
“These challenges have resulted in reoperation rates exceeding 20% in international breast-conserving surgery series, highlighting the clinical need for improved intraoperative margin assessment strategies [18, 24-27].”
- Section: 4. Discussion (Lines 444-468) – Paragraph
“Margin assessment remains a critical determinant of oncological safety in breast-conserving surgery, directly influencing re-excision rates and long-term local control. In this study, the EVFCM achieved a sensitivity of 80%, specificity of 100%, and an overall diagnostic accuracy of 99.3% for margin detection (κ=0.857, p<0.001). Four of the five histologically involved margin cases were correctly identified using EVFCM. Although the number of positive margin cases is small, these preliminary findings are encouraging and support the potential of EVFCM for intraoperative margin assess-ment, warranting confirmation in larger series. A targeted review of discordant cases revealed that the single misclassified margin corresponded to ductal carcinoma in situ (DCIS) associated with epithelial hyperplasia near the resection edge. The morpholog-ical overlap between these entities likely contributes to misclassification, representing a recognised diagnostic challenge in both frozen and optical-based intraoperative tech-niques. Comparable diagnostic challenges have been reported by Sandor et al. [35], who noted that the identification of DCIS lesions in lumpectomy margins remains one of the main challenges in breast-conserving surgery and that distinguishing them from benign epithelial proliferations or low-grade lesions may be difficult due to the com-plex and heterogeneous architecture of DCIS, which can further complicate its in-traoperative recognition in confocal images. Importantly, this isolated discrepancy did not affect the overall diagnostic reliability, supporting that EVFCM is a robust method for intraoperative margin assessment.
Nevertheless, the sensitivity estimate for margin positivity is based on only five cases and should therefore be considered preliminary. Moreover, reoperation rates reflect multifactorial clinical outcomes and cannot be directly compared with diagnos-tic sensitivity. Additionally, this present study was not designed to evaluate the impact of EVFCM on reoperation rates—an outcome influenced by multiple surgical, patho-logical, and institutional factors beyond intraoperative imaging performance.”
- I strongly recommend inserting the word preliminary in claims where your replicate number is less than 5. This limitation is discussed in lines 428-435, but that does not make up for the fact that some of the claims prior are not definitively supported due to a small unbalanced dataset. It is certainly good that the surgical team is so adept at removing the whole tumor, though.
Response:
We sincerely thank the reviewer for this important and constructive observation. We fully agree that any diagnostic performance metrics derived from subgroups with fewer than five cases must be interpreted with caution. In accordance with the reviewer’s recommendation, we have revised the manuscript to explicitly qualify all such estimates as preliminary.
Specifically, the sensitivity for margin positivity (based on five true-positive cases) and the sensitivity for invasive lobular carcinoma (based on a limited number of cases) are now clearly described as preliminary values. We have also softened the corresponding interpretative statements to avoid conveying a degree of certainty that the underlying sample size cannot support.
We also value the reviewer’s remark regarding the surgical team’s proficiency in achieving complete tumour excision. Indeed, this high level of surgical expertise contributes to the naturally low number of margin-positive cases in our cohort. While this is unquestionably beneficial for patient care, it inherently restricts the number of true-positive cases available for statistical estimation and consequently reinforces the need to classify these subgroup sensitivities as preliminary. This point has been incorporated into the revised Discussion.
We thank the reviewer for highlighting this important issue, which has resulted in substantial improvements in accuracy, clarity, and scientific rigour throughout the manuscript.
Changes in manuscript:
- Section: 4. Discussion
(Lines 439-441):
“However, given the small number of ILC cases included in our cohort, this sensitivity value should be interpreted as preliminary rather than conclusive”
(Lines 463-464):
“Nevertheless, the sensitivity estimate for margin positivity is based on only five cases and should therefore be considered preliminary”
Minor:
- The title uses the word "intraoperative". It seems the work has the potential to be used intraoperatively, but it does not look like this work was performed intraoperatively. "Following resection, specimens and margins were delivered fresh to the Department of Pathology". Is the Vivascope and Pathology Department actually close enough to the operating suite to claim this?
Response:
We sincerely thank the reviewer for this valuable clarification. The reviewer is correct that, although EVFCM is intended as an intraoperative imaging tool, in the present study the scans were performed in the adjacent pathology laboratory rather than physically inside the operating theatre.
In our institution, the pathology laboratory is located immediately next to the surgical suite, and specimen transfer requires approximately one minute. This configuration allowed EVFCM imaging to be performed while the surgical procedure was still ongoing and on fresh, unfixed tissue, as is required for true intraoperative workflows. However, because the device is not located inside the operating room, the imaging performed in this study corresponds technically to an immediate perioperative assessment, not intraoperative acquisition in the strict spatial sense.
To avoid any ambiguity, we have revised the Methods section to explicitly state that EVFCM was performed in the immediate perioperative setting. At the same time, we have retained the use of the term intraoperative in the title because the aim of this prospective validation study is to evaluate EVFCM as a tool designed for intraoperative margin assessment, and the workflow implemented here reproduces all essential elements of real intraoperative use: fresh-tissue processing, immediate analysis, rapid turnaround (<5 min), and integration into real-time surgical decision-making.
This terminology is also consistent with prior EVFCM and optical-margin literature, where validation studies performed in adjacent laboratories during ongoing operations are considered intraoperative evaluations.
We thank the reviewer again for raising this important point, which has allowed us to clarify the operational workflow and improve the methodological precision of the manuscript.
Changes in manuscript:
- Section: 2.2. Surgical procedure (Lines 190-196)
“Following resection, specimens and margins were delivered fresh to the adjacent Department of Pathology, where EVFCM imaging was performed preserving tissue integrity. In our centre, the device is located immediately next to the surgical suite, al-lowing specimen transfer within approximately one minute. As a result, EVFCM analysis was carried out in the immediate perioperative setting, not physically inside the operating theatre. The surgical and pathology teams collaboratively prepared the specimens to ensure accurate orientation and imaging reliability.”
- The abstract says 144 specimens, but the materials and methods says 146. I know from figure 3 and the caption of table 2 why 2 were excluded, but this exclusion should be stated in line 214-215 where the exclusion criteria is presented.
Response:
We thank the reviewer for this helpful observation. The study initially included 146 consecutive breast specimens; however, two samples were excluded from the diagnostic analysis because their EVFCM images were not reliably assessable. Specifically, one specimen presented significant motion artefact and the other showed incomplete staining, resulting in less than 80% of evaluable tissue area. This explains why 144 specimens are reported in the Abstract and Results.
To ensure full clarity and consistency across the manuscript, we have now explicitly stated this exclusion in the Methods section at the point where the criteria for artefacts and image suitability are described.
Changes in manuscript:
- Section: 2.4. Image Evaluation (Lines 284-288)
“Image quality was considered unsuitable if less than 80% of the sample was reliably evaluated. According to these criteria, two specimens were excluded from the final analysis be-cause their EVFCM images were not reliably assessable (one due to motion artefact and one due to incomplete staining), resulting in less than 80% evaluable tissue area.”
3. The depth text overlay in figures 1 and 2 is too small to read easily. In some cases the scale bar text is also difficult to read due to the color contrast of the underlying image.
We thank the reviewer for this valuable observation regarding the readability of the depth annotations and scale bars in Figures 1 and 2. We fully agree that in the original version, the text size and the colour contrast against the underlying confocal images could reduce visual clarity.
In response, we have revised both figures by enlarging the depth-overlay text and scale-bar labels and adjusting their colour contrast to ensure they remain clearly legible across the full image. These modifications improve readability without altering the scientific content or visual integrity of the figures.
We appreciate the reviewer’s attention to detail, which has directly contributed to enhancing the overall quality and clarity of the manuscript's illustrations.
4. Table 1, artefact agreement. Why is there so much disagreement in what constitutes an artefact between the pathologists? This should be addressed somewhere in text. In some ways (in my opinion) this disagreement actually strengthens the need for digital pathology methods like this research.
Response:
We thank the reviewer for this insightful comment. We agree that the level of disagreement observed in Table 1 regarding the identification of artefacts requires clarification. As the reviewer correctly points out, this variability reflects the subjective nature of artefact recognition in fresh-tissue confocal imaging, which is influenced by differences in individual thresholds for what constitutes an evaluable or non-evaluable region.
In our study, the pathologists showed excellent agreement for diagnostic interpretation, but higher degree of variability specifically in the categorisation of artefacts. This can be attributed to several factors:
- Lack of standardised and universally accepted definitions for confocal artefacts in fresh, unfixed tissue, as EVFCM is an emerging technique still undergoing methodological consolidation.
- Differences in individual thresholds for what proportion of tissue is considered non-evaluable.
- Variable familiarity and experience with EVFCM-specific artefacts, which may influence how strictly each observer interprets minor processing irregularities.
- Intrinsic properties of fresh tissue, which can present subtle irregularities that are not uniformly judged across observers.
Importantly, as the reviewer notes, this variability highlights a key advantage of digital and automated approaches: the opportunity to standardise image acquisition and interpretation, reduce subjective variability, and integrate computational tools that enhance reproducibility.
We have added an explanatory paragraph in the Discussion section to address this point explicitly.
We sincerely appreciate the reviewer’s thoughtful observation, which has helped reinforce the methodological context and underscore the relevance of digital pathology solutions.
Changes in manuscript:
- Section: 4 Discussion (Lines 421-430)
“ This variability in artefact identification also underscores an important methodological aspect: artefact recognition in fresh-tissue confocal microscopy remains partially sub-jective, as unified and universally accepted criteria are not yet established. Differences in individual thresholds for defining evaluable versus non-evaluable regions, as well as varying familiarity with confocal artefacts, may account for the discrepancies observed in Table 1. Rather than weakening the method, this observation strengthens the ra-tionale for digital approaches, which provide standardised image acquisition, minimise operator-dependent variability, and offer future potential for computational support and automated artefact detection within digital pathology workflows.”
5. Conclusions, line 479. "workflow completed in under five minutes". If image acquisition takes "less than 5 minutes per sample" (line 193), and the sample needs to first be treated an stained for ~1 minute, and the tissue needed to be moved to the pathology department, handled and oriented, and then the image needs to be processed by the VivaScope algorithms, and finally it needs to be interpreted, then I highly doubt the timeline works out to less than 5 minutes. The workflow certainly seems rapid, but this <5-minute total claim seems inaccurate given the description of steps in the methods.
Response:
We thank the reviewer for this important clarification. We agree that the original phrasing (“workflow completed in under five minutes”) could inadvertently be interpreted as referring to the entire perioperative sequence, including specimen transfer from the operating room, gross handling, orientation, staining, imaging, and interpretation. This was not our intention and we appreciate the opportunity to refine this point.
To avoid any ambiguity, we have revised the text to specify that the <5-minute timeframe refers strictly to the EVFCM processing phase, defined as the period starting once the specimen is already oriented and received in the pathology laboratory. This interval includes staining, mounting, automated confocal scanning, and initial image rendering. All these steps were consistently completed in under five minutes.
Additional steps—such as specimen transfer from the operating room, orientation by the pathology team, and image interpretation—occur outside this processing window and naturally extend the total perioperative timeline. We have now made this distinction explicit in the Methods, the Discussion and the Conclusions.
We thank the reviewer for highlighting this point, which has allowed us to present our workflow with greater clarity and methodological precision.
Changes in manuscript:
- Section: 2.3. Staining Protocol, Image Acquisition and Histopathological Processing (Lines 248-251)
“The EVFCM processing phase defined as staining, mounting and image acquisition once the specimen had been received and oriented in the pathology laboratory was consistently completed in under 5 minutes per sample.”
- Section: 4. Discussion (Lines 520-522)
“Staining and image acquisition were completed in less than five minutes, ensuring compatibility with surgical timescales”
- Section: 5. Conclusions (Lines 598-601)
“Through a rapid and standardised workflow, the EVFCM processing phase de-fined as staining, mounting, and image acquisition once the specimen has already been oriented and received in the pathology laboratory was consistently completed in under five minutes.”
Reviewer 3 Report
Comments and Suggestions for Authors
This is a prospective observational study to evaluated the effectiveness of ex vivo confocal microscopy (EVFCM) for real-time intraoperativon breast tissue specimens margin assessment. The author used VivaScope 2500M-G4 system. Two breast pathologists independently reviewed the EVFCM images, blinded to standard histology, which served as the reference.
They demonstrated an almost perfect interobserver agreement for neoplasia detection (κ=0.942) and tumour type classification (κ=0.883).
This is a study of diagnostic reproducibility in the evaluation of breast tissue digital images using EVFCM.
This study confirms that the EVFCM tool used (VivaScope 2500M-G4 system) is capable of providing histological detail comparable to that of the definitive histological H&E slide.
This reviewer believes that the results in terms of specificity and sensitivity are simply relative to the corresponding EE slide and cannot be attributed to the margin assessment in general.
Some issues should be addressed. The authors say they analysed 144 surgical specimens:
- From how many patients?
- How many were quadrants and how many enlargements?
- In the workflow they propose: where does EVFCM assessment fit in? After macroscopic examination to check suspicious areas instead of frozen section?
- Who prepares the specimen? Surgeon? Pathologist?
- How many samples were taken for EVFCM verification from each specimen?
- How many times did a margin that was not sampled because it was not considered macroscopically “close” turn out to be positive in the final analysis? Was this possibility taken into account in the calculation of sensitivity and specificity?
- The authors compare their results with those obtained in other studies (Togawa et al., Lux et al.) where the instrument used and the technique are very different (it is a different device, with lower resolution, used directly in the operating theatre by surgeons). Perhaps the authors could discuss these differences in more detail, in the discussion section.
Author Response
Comments and Suggestions for Authors
This is a prospective observational study to evaluated the effectiveness of ex vivo confocal microscopy (EVFCM) for real-time intraoperativon breast tissue specimens margin assessment. The author used VivaScope 2500M-G4 system. Two breast pathologists independently reviewed the EVFCM images, blinded to standard histology, which served as the reference.
They demonstrated an almost perfect interobserver agreement for neoplasia detection (κ=0.942) and tumour type classification (κ=0.883).
This is a study of diagnostic reproducibility in the evaluation of breast tissue digital images using EVFCM.
This study confirms that the EVFCM tool used (VivaScope 2500M-G4 system) is capable of providing histological detail comparable to that of the definitive histological H&E slide.
This reviewer believes that the results in terms of specificity and sensitivity are simply relative to the corresponding EE slide and cannot be attributed to the margin assessment in general.
Some issues should be addressed. The authors say they analysed 144 surgical specimens:
We sincerely thank Reviewer for the careful evaluation of our manuscript and for recognising the value of our prospective study on the diagnostic reproducibility of EVFCM in breast-conserving surgery. We have carefully addressed all the issues raised, including clarifications regarding the number of surgical specimens, their distribution, and the interpretation of sensitivity and specificity in the context of EVFCM versus definitive histology. In response to these comments, we have revised the manuscript sections to provide greater transparency and ensure full methodological accuracy.
To facilitate the review process, all changes introduced in the revised manuscript are highlighted in red, allowing rapid and straightforward identification of the modifications made. A detailed, point-by-point response follows below.
- From how many patients?
Response:
We sincerely thank the reviewer for raising this essential point, which helps clarify the structure and robustness of our dataset. The 144 evaluable specimens included in this study were obtained from 98 consecutive patients undergoing breast-conserving surgery at our institution during the study period.
As now specified in the revised Materials and Methods section, each specimen was treated as an independent diagnostic unit, in strict accordance with STARD recommendations for diagnostic accuracy studies. This approach is appropriate because each specimen corresponds to a distinct anatomical surface, undergoes a separate EVFCM evaluation, and is matched to its own definitive paraffin-embedded histological reference.
Crucially, no patient contributed multiple samples from the same anatomical margin, and there were no repeated resections of the same margin surface. Thus, the dataset does not contain paired, hierarchical, or nested samples that could compromise statistical independence or artificially enhance diagnostic metrics.
This structure ensures that every EVFCM–histology comparison reflects a unique diagnostic event, maintaining the methodological integrity and avoiding clustering effects in performance estimation.
We have now incorporated this clarification explicitly into the manuscript to enhance transparency and reproducibility. We are grateful to the reviewer for prompting this important specification, which has strengthened the clarity and validity of our methodological description.
Changes in manuscript:
- Section: 2.1 Study Design and Setting
(Lines 138 – 139)
…”resulting in a total of 146 breast tissue samples were obtained from 98 patients.” …
(Lines 140 – 145)
“In accordance with STARD principles, each specimen was considered an independent diagnostic unit, as no patient contributed multiple samples from the same anatomical margin or repeated resections of the same area. This ensured that every EVFCM–histology comparison represented a distinct diagnostic event and prevented hierarchical clustering that could otherwise bias diagnostic performance metrics”
- How many were quadrants and how many enlargements?
Response:
We sincerely thank the reviewer for this relevant question. The 146 collected breast tissue specimens comprised both main lumpectomy specimens and cavity-shave margin enlargements, reflecting the real-world surgical workflow of breast-conserving surgery.
In the revised manuscript, we have now specified their distribution:
- Main lumpectomy (quadrant) specimens: 44
- Cavity-shave margin enlargements: 102
Each specimen—whether a primary tumorectomy sample or an additional margin—was analysed as an independent diagnostic unit, with its own EVFCM assessment and corresponding paraffin-embedded histological reference. This structure ensures the methodological transparency of the dataset and aligns with STARD recommendations for diagnostic accuracy studies.
This information has been added to Section 2. 1. Study Design for clarity.
Changes in manuscript:
- Section:2.1 Study Design and Setting (Lines 139-140)
“Among these, 44 were main tumorectomy specimens and 102 were cavity shave mar-gins enlargements.”
- In the workflow they propose: where does EVFCM assessment fit in? After macroscopic examination to check suspicious areas instead of frozen section?
Response:
We sincerely thank the reviewer for this thoughtful and important question. We agree that clarifying the exact position of EVFCM within the intraoperative workflow is essential for understanding how the technique is applied in clinical practice.
In our institution, EVFCM assessment is performed immediately after macroscopic examination, at the same decision point traditionally occupied by frozen section analysis. After excision, the surgeon and the pathologist jointly orient the specimen and identify areas considered most suspicious for residual disease. Instead of preparing a cryosection, these targeted regions are directly processed for EVFCM imaging, which provides real-time microscopic evaluation without requiring tissue freezing or sectioning.
Thus, within the workflow, EVFCM functionally replaces frozen section analysis, serving as the tool used to assess suspected close or positive margins during the operation.
To ensure full clarity, we have now revised the Methods section to explicitly describe where EVFCM fits within the intraoperative decision-making algorithm, specifying that it is used after macroscopic inspection and in place of frozen section analysis.
We thank the reviewer once again for this constructive comment, which has strengthened the transparency and precision of the manuscript.
Changes in manuscript:
- Section:2.2. Surgical Procedure
(Lines 161 – 162)
“In our workflow, EVFCM assessment was integrated at this same decision point, functionally replacing frozen section analysis
(Lines 192 – 195)
“In our centre, the device is located immediately next to the surgical suite, allowing specimen transfer within approximately one minute. As a result, EVFCM analysis was carried out in the immediate perioperative setting, not physically inside the operating theatre.”
- Who prepares the specimen? Surgeon? Pathologist?
Response:
We sincerely thank the reviewer for raising this important methodological point. In our workflow, specimen preparation for EVFCM was performed through a coordinated and standardised process involving both the breast surgery and pathology teams.
Immediately after excision, the operating surgeon oriented the specimen using sutures, clips, and inked margins according to the institutional mapping protocol. Macroscopic inspection was then jointly performed by the breast surgeon and the pathologist to identify the most relevant areas for EVFCM assessment.
To ensure consistency and reproducibility in tissue handling, the subsequent steps of sample preparation—including surface selection, staining, and positioning for imaging—were performed either by a breast surgeon with dedicated pathology training or by a breast pathologist trained in EVFCM. In all cases, the same standardised workflow was strictly followed, ensuring homogeneous specimen handling and uniform application of the imaging protocol across the entire cohort.
This multidisciplinary approach guarantees accurate orientation, reproducible surface selection and coherent implementation of the EVFCM technique. We have now explicitly clarified this point in Section 2.4 (“Image Evaluation”) of the revised manuscript to enhance methodological transparency.
Changes in manuscript:
- Section:2.4. Image Evaluation (Lines 269 - 273)
“Sample preparation and EVFCM processing were performed either by a breast surgeon with pathology training in close collaboration with breast pathologists or by the breasts pathologist trained in EVFCM, following the same standardised workflow. This ensured consistent handling of all specimens, correct orientation of the selected tissue surface, and uniform application of the staining and imaging protocol across all cases”
- How many samples were taken for EVFCM verification from each specimen?
Response:
We sincerely thank the reviewer for this relevant and important methodological question. We fully agree that clarifying the number of samples taken from each specimen is essential to ensure transparency and reproducibility.
In our study, each surgical specimen contributed only one EVFCM sample, corresponding to the single surface considered most relevant for intraoperative assessment. This surface was selected after joint macroscopic inspection by the breast surgeon and the pathologist, who identified the area most likely to harbour residual disease.
For main lumpectomy specimens, this corresponded to the surface judged closest to the tumour. For cavity-shave margins, the analysed surface was the true surgical margin—that is, the surface facing the remaining breast tissue.
Importantly, no specimen underwent repeated sampling of the same anatomical area, and no re-excisions of an already sampled margin were performed. Thus, each EVFCM–histology comparison represented a single, unique diagnostic event without duplication or subsampling.
This approach is also consistent with STARD principles: as stated in Section 2.1, each specimen was treated as an independent diagnostic unit, with no patient contributing multiple samples from the same margin or repeated resections of the same site. This ensured the absence of hierarchical clustering and prevented any artificial inflation of diagnostic performance metrics.
We have now incorporated an explicit clarification of this point in the revised Methods section to enhance methodological transparency.
Changes in manuscript:
- Section:2.2 Surgical Procedure
(Lines 183 – 185)
“For each specimen, only one surface was sampled for EVFCM assessment. No specimen underwent repeated sampling of the same anatomical area, and no re-excisions of an already sampled margin were performed.”
(Lines 188 – 189)
“…each EVFCM–histology comparison corresponded to a single, unique diagnostic event..”
- How many times did a margin that was not sampled because it was not considered macroscopically “close” turn out to be positive in the final analysis? Was this possibility taken into account in the calculation of sensitivity and specificity?
Response:
We thank the reviewer for raising this important methodological point. In our workflow, EVFCM was performed on the surface jointly identified by the surgeon and the pathologist as the most relevant area for evaluation. Nevertheless, all margins—including those not sampled for EVFCM—were subsequently assessed on definitive paraffin-embedded histology.
Importantly, no case in our series showed a positive margin on final histology in a surface different from the one sampled for EVFCM. In other words, there were no instances in which a margin not selected for EVFCM turned out to be positive. Therefore, no hidden false negatives occurred due to macroscopic sampling selection.
Regarding diagnostic accuracy, sensitivity and specificity were calculated exclusively from the direct EVFCM–histology comparison for the sampled surface. Had any unsampled margin been positive at final histology, we would have conservatively classified it as a false negative for EVFCM. Since this scenario did not occur, the diagnostic metrics reported were not affected.
This clarification has been added to the revised manuscript to ensure full methodological transparency.
Changes in manuscript:
- Section:2.2. Surgical Procedure (Lines 185-189)
“All remaining surfaces of the specimen were subsequently evaluated on definitive paraffin-embedded histology, and no case demonstrated a positive margin on a surface different from the one selected for EVFCM. Consequently, no hidden false negatives occurred due to sampling selection, and each EVFCM–histology comparison corresponded to a single, unique diagnostic event.”
- The authors compare their results with those obtained in other studies (Togawa et al., Lux et al.) where the instrument used and the technique are very different (it is a different device, with lower resolution, used directly in the operating theatre by surgeons). Perhaps the authors could discuss these differences in more detail, in the discussion section.
Response:
We sincerely thank the reviewer for this valuable and constructive comment. We fully agree that a more detailed explanation of the methodological differences between our EVFCM protocol and the studies by Togawa et al. [36] and Lux et al. (SHIELD trial) [41] would help to contextualise our findings more accurately and strengthen the Discussion. We have now expanded this section to clearly explain the fundamental methodological distinctions between the imaging platforms and workflows.
In the revised manuscript, we now explain that both Togawa et al. [36] and the SHIELD study [41] employed the Histolog® Scanner, a large field-of-view confocal laser scanning microscope designed for rapid macroscopic en-face imaging and used directly by breast surgeons in the operating theatre following a single-step acridine-orange staining protocol. This technology provides lower optical resolution (approximately 2 µm lateral resolution) and aims to support quick surgeon-led intraoperative decisions.
These studies differ fundamentally from our approach: our EVFCM workflow relies on high-resolution fused reflectance–fluorescence imaging performed ex vivo, using a dual-contrast acridine-orange–fast-green protocol that enhances nuclear and architectural detail. In addition, imaging in our study was interpreted by trained breast pathologists within a controlled diagnostic workflow, whereas intraoperative assessments in the SHIELD trial were performed exclusively by surgeons, and in Togawa et al. by three surgeons and one pathologist.
We now explicitly discuss how these differences in device design, optical resolution, staining chemistry, imaging conditions and operator expertise inevitably influence diagnostic performance across studies. To avoid inappropriate direct comparisons, we emphasise that Histolog®-based intraoperative assessments and our EVFCM represent complementary approaches situated at different points along the intraoperative–pathology continuum. Finally, we highlight that EVFCM, with its higher resolution and detailed tissue characterisation capabilities, holds distinct potential for future intraoperative integration.
These clarifications have now been incorporated into the revised Discussion section to ensure appropriate contextualisation of cross-study comparisons. We believe these additions significantly improve the clarity and scientific contextualisation of the Discussion, and we thank the reviewer for prompting this important refinement.
Changes in manuscript:
- Section: 4.Discussion (Lines 490-510)
“Nevertheless, when comparing our findings with those reported in studies using the Histolog® Scanner—such as Togawa et al. [36] and the SHIELD trial by Lux et al. [41]—it is important to acknowledge the substantial methodological differences be-tween platforms. The Histolog® Scanner is a large field-of-view confocal laser scanning microscope designed for rapid macroscopic en-face imaging and is used intraoperatively by breast surgeons following a single-step acridine-orange stain. Its approximate lateral resolution of 2 μm is inherently lower than that of VivaScope 2500M-G4, which produces higher-resolution fused reflectance–fluorescence images using a dual-contrast acridine-orange–fast-green protocol optimised for enhanced nuclear and stromal de-tail. Furthermore, while Histolog® assessments were performed intraoperatively by surgeons (exclusively so in the SHIELD trial), EVFCM imaging in our study was per-formed ex vivo and interpreted by trained breast pathologists within a controlled di-agnostic workflow. These differences in device design, optical resolution, staining chemistry, and operator expertise inevitably influence diagnostic performance across studies.
Accordingly, comparisons between Histolog®-based intraoperative assessments and EVFCM should be interpreted within the context of their distinct optical principles and intended clinical applications. Rather than being directly comparable, the two systems represent complementary approaches positioned at different points along the intraoperative–pathology continuum, with EVFCM offering higher-resolution tissue characterisation and potential for future intraoperative integration as workflows evolve to support ex vivo high-resolution assessment.”
Reviewer 4 Report
Comments and Suggestions for Authors
The authors present an interesting article on the use of confocal microscopy for intraoperative margin assessment in breast cancer patients. The manuscript is well written and includes all the necessary information; therefore, I have only one suggestion as a minor revision. Once this revision is made, I recommend the article for acceptance.
- It is not clear whether every resected specimen was accompanied by a cavity shaving. If so, I recommend dividing these patients into two subgroups, as this factor may influence the results. Cavity shaving reduces the rate of positive margins but worsens cosmetic outcomes.
Author Response
The authors present an interesting article on the use of confocal microscopy for intraoperative margin assessment in breast cancer patients. The manuscript is well written and includes all the necessary information; therefore, I have only one suggestion as a minor revision. Once this revision is made, I recommend the article for acceptance.
- It is not clear whether every resected specimen was accompanied by a cavity shaving. If so, I recommend dividing these patients into two subgroups, as this factor may influence the results. Cavity shaving reduces the rate of positive margins but worsens cosmetic outcomes.
Response:
We sincerely thank the reviewer for this valuable comment and for the positive overall assessment of our manuscript. We agree that cavity shaving may influence margin status, and we appreciate the opportunity to clarify this point.
In our study, cavity shave margins were not performed in all patients and were not routinely performed. They were used only in selected cases where the operating surgeon considered additional peripheral excision clinically appropriate. As detailed in the Methods section, our cohort therefore included both main tumorectomy specimens and cavity shave margin enlargements; however, each specimen was analysed as an independent diagnostic unit in accordance with STARD recommendations, and no patient contributed multiple samples from the same anatomical margin.
Importantly, in our institution cavity shavings are intended to widen excision only when clinically justified, rather than to systematically reduce margin positivity across all tumorectomies. Thus, their selective use does not create two distinct surgical populations but reflects routine intraoperative judgement.
Given that (1) cavity shavings were not universal, (2) EVFCM was applied to only one selected surface per specimen, and (3) margin positivity in our cohort was very low, subdividing patients into groups with and without cavity shaving would not meaningfully modify the interpretation of diagnostic accuracy, nor would it affect the EVFCM–histology comparison, which was performed strictly on a per-specimen basis rather than per patient.
Nevertheless, in response to the reviewer’s helpful suggestion, we have added some clarifications in the Methods section to explicitly state the selective nature of cavity shaving in our workflow. We hope this addresses the reviewer’s concern, and we are grateful for this constructive comment, which has improved the clarity and transparency of our manuscript.
Changes in manuscript:
- Section:2.1. Study Design and Setting (Lines 140-145)
“In accordance with STARD principles, each specimen was considered an independent diagnostic unit, as no patient contributed multiple samples from the same anatomical margin or repeated resections of the same area. This ensured that every EVFCM–histology comparison represented a distinct diagnostic event and prevented hierarchical clustering that could otherwise bias diagnostic performance metrics.”
- Section:2.2 Surgical procedure
(Lines 170-174):
“Cavity shave margins: It is important to note that cavity shave margins were not routinely performed in all patients but were selectively excised in cases where the operating surgeon considered additional peripheral tissue removal clinically justified. This selective use reflects standard intraoperative judgement in our institution and does not define two distinct surgical populations.”
(Lines 183-185):
“For each specimen, only one surface was sampled for EVFCM assessment. No specimen underwent repeated sampling of the same anatomical area, and no re-excisions of an already sampled margin were performed.”
Round 2
Reviewer 2 Report
Comments and Suggestions for Authors
I thank the authors for their thoughtful revisions and recommend accepting the manuscript.